# Scenario-based forecast of the evolution of 75 years of unrest at Campi Flegrei caldera (Italy)
Luca Caricchi [1] ✉, Charline Lormand[1], Stefano Carlino [2], Tommaso Pivetta[2] & Guy Simpson[1]

Campi Flegrei last erupted in 1538 and periods of increased seismicity, gas emission and ground deformation occurred in the 50's, 70's 80's and are ongoing since 2005. The eventual culmination of the unrest in an eruption, would directly impact on 2 million people living in the region, making it of critical concern for scientists, authorities and the public. Here, we use existing data, thermal modelling and calculations of the physical properties of magma, to provide plausible future scenarios, under the assumption that magma injection at 4-5 km depth is responsible for the unrest episodes recorded since 1950. Our calculations suggest that a critically pressurised reservoir containing potentially eruptible magma is present today at ~ 4 km depth. However, a major impediment to eruption is the reservoir volume, which would need 2-3 decades to grow to the size of the one that fed the last eruption of Campi Flegrei in 1538.

In the last 75 years, the Campi Flegrei caldera (CFc) has experienced four episodes of unrest in 1950–52, 1970–72, 1982–84, and 2005-present[1,2]. The surface uplift during these episodes was 74, 160, 175 cm, respectively, and is larger than 130 cm at the time of writing (refs. [3,4]). Between 1984 and 2005, the caldera floor subsided about 90 cm, thus recovering about half of the inflation measured in the crisis of 1982–1984 (ref. [1]). Now, the uplift has already exceeded the values reached in the 1982–1984 crisis and is therefore at the maximum since at least 1950. The mechanism currently generating inflation is still debated, with one possibility being the injection of magma at depth corresponding to the estimated depth of the inflating source (4–5 km; refs. [5–7]) and another being an increased flux of magmatic fluids released from deeper within the volcanic plumbing system[8–11]. Geodetic data for the 1982–1984 and the ongoing unrest episodes are best explained by volume increase of a source located at 4–5 km depth of $64–87 \times 10^6$ and $60 \times 10^6 \, \text{m}^3$, respectively[6,12]. Volumetric estimates for the sources of the 1950 and 1970 crises do not exist, thus we assume a proportionality between the total measured uplift and the corresponding increase of volume estimated for the last two episodes (Supplementary Table S1), and a similarity in the depth of the source of deformation[13].

## Results and discussion

Using the volumes obtained from geodetic inversion and estimated for the unrest episodes of the 50's and 70's, we simulate an end-member extreme scenario where we assume that each inflation episode is caused by magma injection. We perform thermal modelling simulating the injection of magma at 4 km depth to trace the evolution of temperature, crystallinity, the

capacity of magma to erupt (eruptibility; magma is more likely to erupt if it contains less than 50 vol% crystals; ref. [14]), excess fluid fraction and magma compressibility, within this shallow reservoir from 1950 to today (Methods; Supplementary Tables S1, S2). Our simulations do not aim to reproduce the complex evolution of the unrest at CFc over the last 75 years, but to provide first-order estimates of the physical state of magma potentially present at shallow depth and to trace the internal overpressure in this shallow reservoir. We also consider the deflation between 1984 and 2005[15,16] to verify if our calculations are appropriate, considering that the calculated volume of excess fluids present within the reservoir in this time interval should be at least sufficient to account for the volumetric decrease responsible for the deflation. As we do not consider any poroelastic contribution of the hydrothermal system to deflation and we ascribe it exclusively to the release of magmatic fluids, the volume of excess fluids required to account for the deflation observed between 1984 and 2005 is possibly smaller than the values obtained from our calculations.

### Temporal evolution of crystallinity, volatile content and physical properties of magma injected in the upper crust

The CFc magma contains up to 4 wt.% of $H_2O$ (refs. [17,18]) and becomes fluid-saturated at about 8 km depth (~200 MPa), with variations related to $CO_2$ content of magma and excess fluids[18]. This depth corresponds to the roof of the main magma reservoir, which develops mostly at depth >8 km (refs. [6,19,20]). Magma ascent from 8 km to the depth at which the current deformation source is located (4 km equivalent to ~100 MPa; refs. [6,20]), results in the degassing of about 1 wt.% $H_2O$ followed by the release of

[1]Department of Earth Sciences, University of Geneva, Geneva, Switzerland. [2]INGV-Sezione di Napoli, Osservatorio Vesuviano, Napoli, Italy. ✉e-mail: luca.caricchi@unige.ch

additional water during cooling and crystallisation (Fig. 1a). This implies that eventual magma accumulating at 4 km depth is fluid saturated. The presence of an excess fluid phase increases magma compressibility (Fig. 1b), which in turn, dampens the increase of overpressure within a magma reservoir caused by further exsolution of volatiles or the injection of additional magma[21,22].

We have performed calculations considering the injection of magma in sill-like intrusions with volumes corresponding to those reported in Supplementary Table S1. The diameter of the injections was adjusted to be

consistent with geodetic inversions[6] and to simulate two different thickness (15 and 25 m; close to the estimates obtained from the inversion of geodetic data[6]; see Methods) for the magma injections as most of the heat is lost along the vertical direction[23], which impacts on the volume of eruptible magma present within the reservoir. Our results show that if magma was injected during each unrest episode that occurred over the last 75 years, the time interval between injections is not sufficient to allow for the temperature to return to the pre-1950's values and the volume of magma above the solidus (e.g. temperature between 850 and 900 °C; Fig. 1) temperature increases

**Fig. 1 | Relationships between magma temperature and melt fraction, mass of exsolved fluids, and compressibility.** Variation of melt fraction, excess fluid fraction (**a**) and magma compressibility (**b**) as function of temperature. These relationships were used to compute the temporal evolution of these parameters in time from the thermal modelling results.

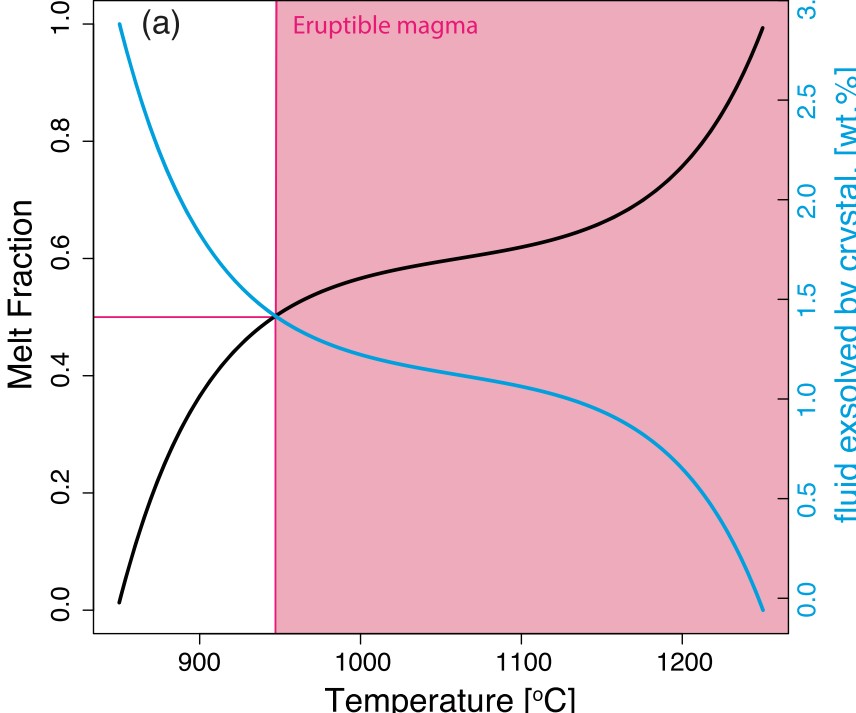

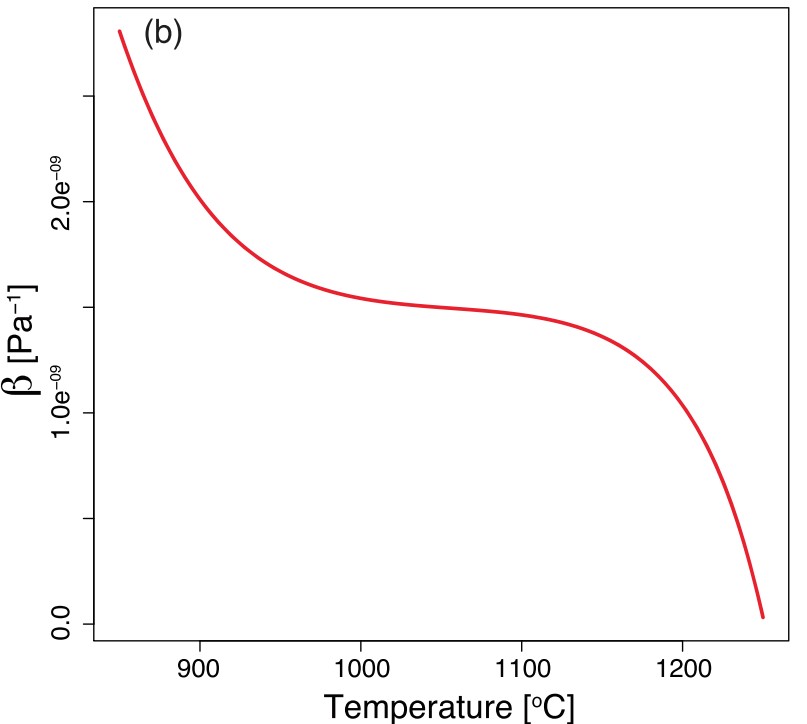

**Fig. 2 | Temporal evolution of the volume of magma at different temperatures and excess fluids.** The figure reports the results for the injections of 25 m thickness, and the maximum and minimum volume of the magma injection events (Supplementary Table S1). The lower end of each coloured region is for the calculations performed considering the minimum volume of injected magma at each event. The numbers close to the dashed lines indicate the minimum and maximum volume of magma injected at each episode. The bands of different colour represent the volumes of excess fluids (in blue and associated with the right-hand y-axis), and magma within different temperatures present within the magma reservoir over time.

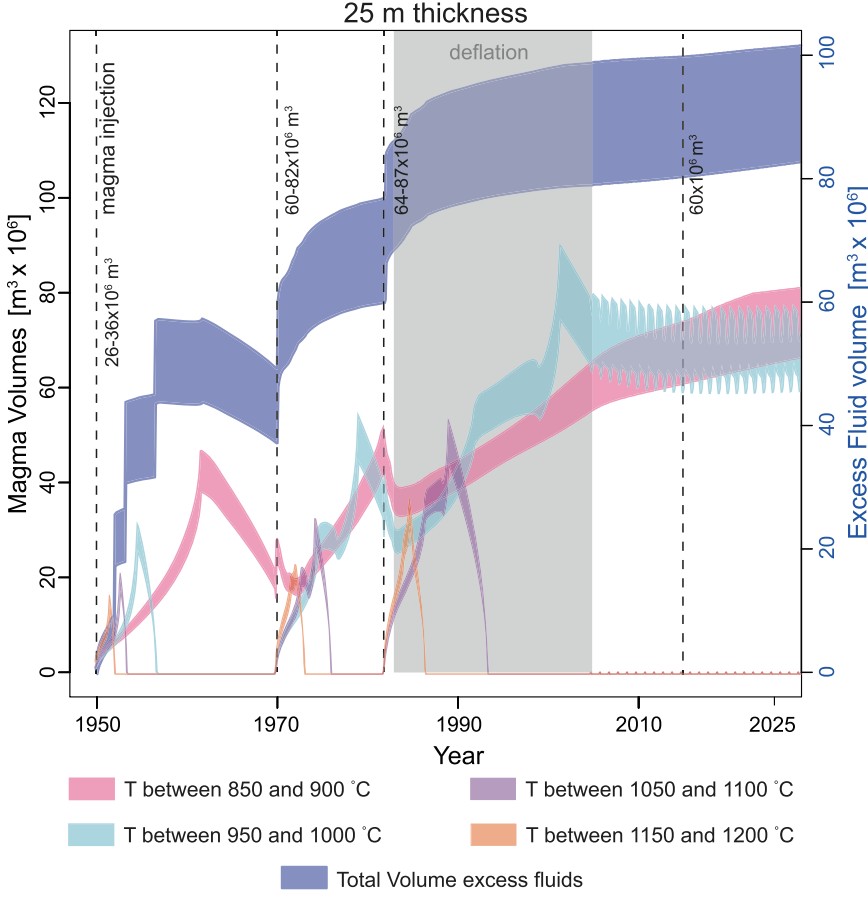

with time (Fig. 2 and Supplementary Fig. S1). The interaction between the rate of magma input, cooling rate, and the release of latent heat of crystallisation, modulates the volumes of magma within the different temperature ranges (Fig. 2 and Supplementary Fig. S1). Importantly, our calculations show that between $60 \times 10^6$ and $80 \times 10^6$ m³ of eruptible magma are present today at 4 km depth, considering a thickness of the injections of 25 m, while almost no eruptible magma would be present if the thickness of the single injections is 15 m (Fig. 2, Supplementary Figs. S1, S3 and Supplementary Table S2). In agreement with existing thermal modelling results[23], this implies that for thicknesses larger than 25 m, the volume of eruptible magma present today would be larger.

The cumulative volume of excess fluids in the reservoir at 4 km depth, and therefore magma compressibility, also increases progressively (Fig. 2; Supplementary Table S2). The volume of excess fluids can be considered in relation to the deflation episode that took place following the 1982–1984 crisis and lasted until 2005. In the period between 1984 and 2005 the areas that uplifted by 175 cm in the 1982–1984 unrest episode, underwent a deflation of 93 cm, thus recovering about half of the inflation. The increase of volume for the 1982–1984 obtained from geodetic inversion was estimated to range between $64 \times 10^6$ and $87 \times 10^6$ m³ (ref. 12). Thus, plausibly, the volume loss from 1984 to 2005 should be half of the volume generating the inflation in 1982–1984 (i.e. $30–45 \times 10^6$ m³). Our calculations show that in the period corresponding to the deflation, about $70–95 \times 10^6$ m³ of excess fluids would be present within the reservoir. These calculations consider the reservoir at 4 km depth as a closed system, which is unlikely as also suggested by the deflation observed after the 1950 and 1970 unrest episodes[1]. Even if half of the volume of excess fluids we obtain from the closed-system degassing calculations, would have been released in the period 1950–1983, the excess fluid volume in the shallow reservoir would still be sufficient to explain the deflation episode independently of the thickness of the injections (Fig. 2, Supplementary Fig. S1), which indicates that our calculations are

correct at least within an order of magnitude. While our calculations account for geodetic data, the chemistry of gas emissions is best explained with the combined contribution of a shallow and a deeper source for the gases[24–26]. Especially, relevant in this respect is the currently observed increase of $CO_2$/$H_2O$ ratio of the gas emissions[11], which cannot be accounted for by crystallisation-induced release of excess fluids[27]. Melt inclusion studies[28], together with the lower solubility of $CO_2$ with respect to $H_2O$ (e.g. ref. 29), suggest that the fluids released from deep within the plumbing system are $CO_2$-rich. The flushing of $CO_2$-rich fluids from depth has two major effects on magma stored at shallow depth: (1) it leads to the exsolution of $H_2O$-rich fluids and (2) it leads to magma crystallisation[30]. As we did not consider this process in our calculations, the volume of excess fluids released by the shallow reservoir could be larger than that obtained from our calculations. Additionally, the crystallisation occurring because of the interaction between $CO_2$-rich fluids and magma at shallow depth would decrease the volume of eruptible magma with respect to our calculations. Finally, as showed by ref. 6, while the contribution of fluids from depths >8 km is required to account for gas monitoring data, the expansion of the deeply sourced fluids, without magma transfer would not be sufficient alone to explain the geodetic data, which suggests that magma accumulation at about 4 km deep is a requirement to account for the geodetic data.

## Mechanisms of reservoir's overpressurisation and temporal evolution of overpressure

Magma injection in a viscoelastic crust results in an increase of overpressure within the magma reservoir, which is proportional to the rate of magma input and the viscosity of the crust, and inversely proportional to the size of the reservoir and magma compressibility[31,32]. In this section, we focus on the current unrest and perform thermal modelling to simulate the injection of magma at $8 \times 10^6$ m³/y, modulated in injections every 0.1 years, starting from 2015. In these simulations magma is injected starting the thermal

structure obtained by running the model from 1950 to 2015. The selected rate is at the high end of the inverted volumetric inflation rate best explaining the inversion of the geodetic measurements[6] to obtain maximum estimates of the overpressure generated by magma injection. Out thermal modelling results allow us to estimate the temporal evolution of the volume of the magma reservoir, its compressibility, and the volume of eruptible magma. In turn, these parameters allow us to determine whether the overpressure within the magma reservoir is sufficiently large to lead to the fracturing of the rocks surrounding the reservoir and allow for the propagation of the magma toward the surface. Mechanical models of reservoir pressurisation in viscoelastic media have highlighted that three timescales control whether a magma reservoir can be pressurised sufficiently to lead to the fracturing of the rocks surrounding the reservoir[33]:

$$\tau_{in} = \frac{\rho V}{\dot{M}_{in}} \qquad (1)$$

$$\tau_{cool} = \frac{R^2}{\kappa} \qquad (2)$$

$$\tau_{relax} = \frac{\eta_r}{\Delta P_c} \qquad (3)$$

Where $\tau_{in}$ is the timescale of magma injection, $\rho$ is the bulk magma density, $V$ is the volume of the magma reservoir, $\dot{M}_{in}$ is the rate of mass injection in the reservoir, $\tau_{cool}$ is the cooling timescale, $R$ is the radius of the reservoir (assumed spherical), $\kappa$ is the thermal diffusivity, $\tau_{relax}$ is the Maxwell relaxation timescale of the crust, $\eta_r$ is the viscosity of the rocks surrounding the reservoir, and $\Delta P_c$ is the critical overpressure required for an eruption to occur. We use the volumes of the reservoir obtained from the thermal model as $V$ (increasing from 240 to $250 \times 10^6$ m$^3$ in 10 years of magma injection) and compute the equivalent radius ($R$) that the reservoir would have if it was spherical. We assume a bulk magma density of 2500 kg/m$^3$ ($\rho$), a $\Delta P_c$ of 10 MPa, a $\kappa$ value of $1.2 \times 10^{-6}$ m$^2$/s, and a crust viscosity ($\eta_r$) of $10^{17}$ Pa s to recalculate these timescales. The results show that the rate of magma input estimated from geodetic data is sufficient to pressurise the reservoir to critical values compatible with the fracturing of the surrounding rocks (Fig. 3). Taking into account the presence of excess fluids (by decreasing $\rho$ from 2500 to 2200 kg/m$^3$), does not affect significantly our calculations[32]. Considering a non-spherical magma reservoir implies an increase of the Maxwell relaxation time of more than one order of magnitude (e.g. refs. 34,35), which will result in an increase of the $\theta_2$. Considering a higher values for $\eta_r$, would decrease $\tau_{relax}$ and thus compensate the effect related to a non-spherical reservoir (Fig. 3). The lowest value of $\theta_2$ would be achieved for a spherical reservoir and a $\eta_r$ one order of magnitude lower than the one considered here. Even under these extreme assumptions, the conclusion that the reservoir is critically over-pressurised does not change (Fig. 3), which agrees with recent findings obtained from the analysis of pattern of seismicity since 2019 (ref. 36).

Our calculations suggest that the overpressure generated by magma input over the last 10 years exceeds the tensile strength of the crust (e.g. ref. 37), which might result in the propagation of dykes accompanied by the depressurisation of the reservoir. The rate of drop in overpressure of the reservoir is inversely proportional to the size of the reservoir and inversely related to magma compressibility. Together these considerations imply that even if a magma reservoir is highly over-pressurised, it might be too small or not enough compressible to feed a volcanic eruption[38,39]. Our thermal modelling results show that the volume of the magma reservoir and of the eruptible magma depends strongly on the thickness of the injected sills and decrease with decreasing thickness (Fig. 2 and Supplementary Fig. S1). The calculated reservoir volume to 2025 varies between $100 \times 10^6$ and about $230 \times 10^6$ m$^3$ (Supplementary Table S2; Fig. S3). Our results show that the average excess fluid fraction at the end of our calculations reaches a value of about 0.22 (Fig. 4a and Supplementary Table S2; calculated for a thickness of the injected sills of 15 m and the minimum injected volume, which

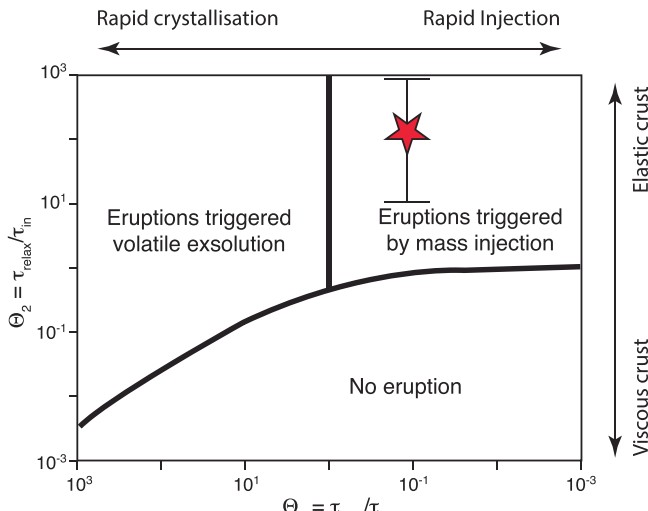

Fig. 3 | Regime diagram showing the dominant timescales associated with different eruption trigger mechanisms. The red star shows the result of the calculations performed on the base of the thermal modelling results presented here. This indicates that the reservoir is potentially sufficiently over-pressurised to lead to the fracturing of the rocks surrounding the reservoir. The black bars indicate the range of uncertainty resulting from an increase of relaxation time for non-spherical reservoirs (increase of $\theta_2$) or associated with lower viscosity of the crust with respect to the value considered here. Variations of the parameters used to calculate the relaxation timescales presented in Eqs. (1)–(3), do not result in a significant change of our result.

maximises the fraction of excess fluid released because of the highest rates of cooling of these low aspect ratio sills). The release of half of the excess fluids from 1950 to today, results in a drop of compressibility from about $1.7 \times 10^{-9}$ to $1.2 \times 10^{-9}$ Pa$^{-1}$ (Fig. 4).

Our thermal model results provide estimates of the current volume of the magma reservoir that vary between 0.10 and 0.23 km$^3$ (Supplementary Table S2), which is about half of the size of the volume of the reservoir required to feed an eruption of the magnitude of Monte Nuovo in 1538 for a reservoir at a. depth of 4 km and a magma water content of 3–4 wt.% (ref. 38). This implies that the accumulation of magma in the reservoir at 4 km depth should continue to reach conditions compatible with an eruption. At a maximum rate of $8 \times 10^6$ m$^3$/y estimated from the inversion of geodetic data collected in the current period of unrest[6], the reservoir should reach conditions compatible with eruption in the next two decades. In the last 50 years, episodes of heightened overpressure could have led to fracturing of the host rock, propagation of magma-filled fractures[38], which, however, did not reach the surface. The drop in reservoir overpressure resulting from the propagation of magma-filled cracks could be responsible to the pauses observed in the surface uplift since 2005 (refs. 6,40).

The calculations we performed are based on an end-member plausible scenario that each volcanic unrest since the 50's was associated with the transfer of magma in the shallow crust. While this scenario is not necessarily valid, it can explain the deflation between 1984 and 2005 by the degassing of magma accumulated at 4 km depth since the 50's, and it allows us to identify critical factors that should be considered to maximise our capacity of anticipating a potential future eruption.

Our results indicate that if all unrest episodes since 1950 were associated with the injection of magma at shallow depth, more than $100 \times 10^6$ m$^3$ of eruptible magma could be present today at about 4 km depth. This is a crucial result, as the presence of eruptible magma is a condition *sine qua non* for a volcanic eruption to occur. Nevertheless, the low viscosity of the crust consequence of the high temperature present at depth below the CFc, magma compressibility, as well as the relatively small size of the reservoir at shallow depth, reduce the likelihood of an internally triggered volcanic eruption. At the current rate of magma injection[6], a couple of decades of

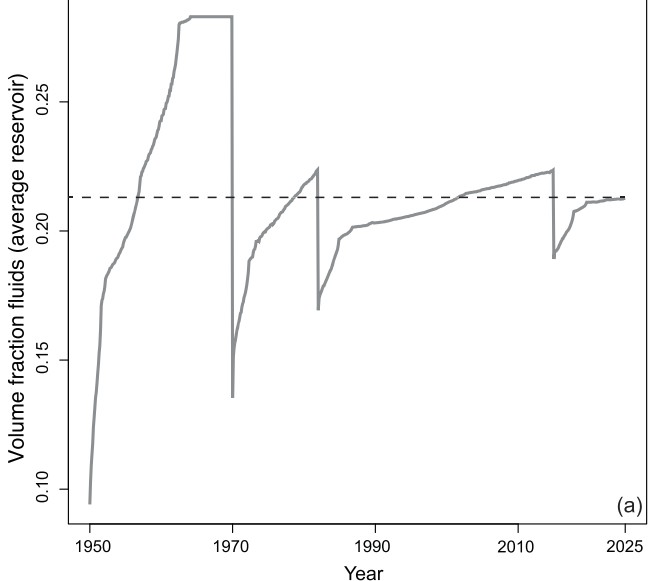

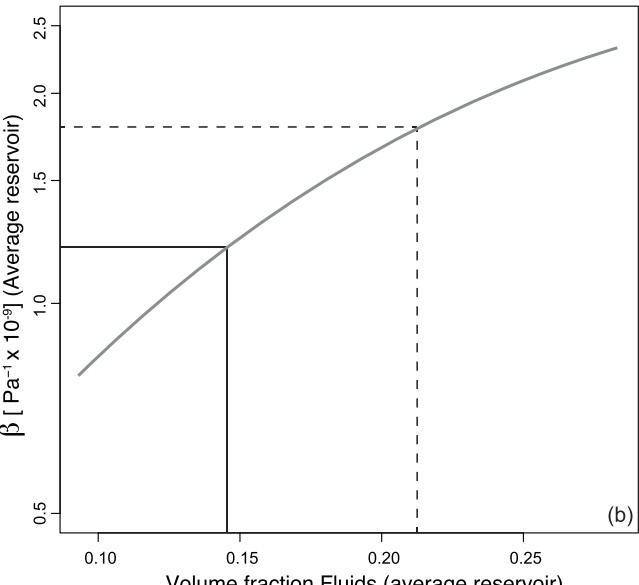

**Fig. 4 | Temporal evolution of the volume fraction of excess fluids and its relationship with magma compressibility. a** Evolution in time of the average fraction of excess fluids present within the magma reservoir as calculated from the thermal model performed considering sills of 15 m thickness and a volume of injected magma corresponding to the minimum estimated volume injection for each episode. **b** Relationships between the average compressibility of magma within the reservoir and the average volume fraction of excess fluids present within the reservoir. The dashed line in panels (**a**) and (**b**) shows the value of compressibility calculated at the end of the simulation. The black line in panel (**b**) shows the drop of compressibility resulting by the removal of about 50% of the exsolved fluids.

magma accumulation would still be required for the conditions to be appropriate for an internally triggered volcanic eruption[38]. Critically, an eruption could be triggered by the propagation of fractures through the 4 km of crust separating the magma reservoir from the surface, stemming from the loss of the integrity of the crust caused by 75 years of volcanic unrest, deformation, and alteration resulting from the circulation of hydrothermal fluids[41,42]. Recent studies focusing on seismicity at Campi Flegrei[43,44] show that since at least 2022, focal mechanisms of earthquakes are becoming more coherent and a portion of a ring fracture is clearly defined by seismicity shallower than 3.7 km, which all together suggest the development of extended fractures in the crust above the reservoir. Thus, we suggest that

monitoring the development of fractures that could ultimately connect the magma reservoir to the hydrothermal system or the surface is of outmost importance, because a well-developed fracture system could facilitate the rise of magma to the surface even if the reservoir volume is not ideal for a volcanic eruption to occur.

## Methods

We numerically compute the thermal evolution of a magma system by solving the heat equation for a quasi-3d (axisymmetric) geometry and a finite elements approach. Our model considers punctual injection of a series of thin cylindrical horizontal sills of constant thickness and variable lateral extent along with heat conduction and release of latent heat. Sills are injected at a constant depth while the material below each sill is advected downward. Boundary conditions were fixed temperature at the base and surface and no horizontal heat flux across the lateral boundaries. Initially, a constant temperature gradient in the vertical direction was assumed.

We simulate the injection of magma at its liquidus temperature of 1250 °C (Fig. 1a; ref. 45) at 4 km depth and a temperature of the rocks surrounding the magma of 540 °C, considering a thermal gradient of 135 °C/km, which is conservative considering the values measured in borehole down to about 3 km depth within the Campi Flegrei caldera, reaching 200 °C/km (ref. 46). We perform the calculations for sills of 15 m and 25 m thickness adjusting the footprint of the sills (considered cylindrical) to match the minimum and maximum volume estimated for each episode of magma injection (Supplementary Table S1).

We use thermodynamic calculations performed at 150–300 MPa to obtain a relationship between temperature and the melt fraction of the magma capturing the salient features of the topology of the phase diagram (Supplementary Fig. S1), which shows two marked decreases of the melt fraction for temperatures from the liquidus (1250 °C) to about 1150 °C and a second for temperatures lower than about 950 °C (Fig. 1). We also parametrised the relationships between temperature and molar volume of excess fluids at a fix pressure of 100 MPa (Supplementary Fig. S2). The values were obtained using the equation of ref. 47. More information about the input parameters of the model is reported in Supplementary Table S3.

## Data availability

The modelling results are stored and publicly available at: https://zenodo.org/records/17791732.

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

## Acknowledgements

This work was supported by the Swiss National Science Foundation (grant no. CRSII5_216582 to L.C.).

## Author contributions

L.C. and S.C. conceived the study; G.S. designed the numerical modelling; C.L. and G.S. performed the simulations; All authors, L.C., C.L., S.C., T.P., and G.S., contributed to result interpretation and the writing of the manuscript.

## Competing interests

The authors declare no competing interests.
