## [Transparent Peer Review file · Communications Earth & Environment]

Scenario-based forecast of the evolution of 75 years of unrest at Campi Flegrei caldera (Italy)

Corresponding Author: Professor Luca Caricchi

Version 0:

Decision Letter:

Dear Professor Caricchi,

Your manuscript titled "Scenario-based and physics-informed forecast of the evolution of 75 years of unrest at Campi Flegrei caldera (Italy)" has now been seen by 4 reviewers, and we include their comments at the end of this message. They find your work of interest, but some important points are raised. We are interested in the possibility of publishing your study in Communications Earth & Environment, but would like to consider your responses to these concerns and assess a revised manuscript before we make a final decision on publication.

We therefore invite you to revise and resubmit your manuscript, along with a point-by-point response that takes into account the points raised, such as the need to provide further details on your methodology and the assumptions made. Please highlight all changes in the manuscript text file.

Please submit your point-by-point responses as a separate file, distinct from your cover letter where you can add responses to the Editors' comments that you do not want to be made available to the reviewers. Word files are preferred. We recommend that any figures, tables or graphs that are included in the response to reviewers are also included in the main article or Supplementary Information.

Please use the following link to submit your revised manuscript, point-by-point response to the referees' comments (which should be in a separate document to any cover letter), a tracked-changes version of the manuscript (as a PDF file) and the completed checklist:

Link Redacted

We hope to receive your revised paper within six weeks; please let us know if you aren't able to submit it within this time so that we can discuss how best to proceed. If we don't hear from you, and the revision process takes significantly longer, we may close your file. In this event, we will still be happy to reconsider your paper at a later date, as long as nothing similar has been accepted for publication at Communications Earth & Environment or published elsewhere in the meantime.

Please do not hesitate to contact us if you have any questions or would like to discuss these revisions further. We look forward to seeing the revised manuscript and thank you for the opportunity to review your work.

Best regards,

Carolina Ortiz Guerrero, Ph.D.
Associate Editor, Communications Earth & Environment

EDITORIAL POLICIES AND FORMATTING

- Behavioural and social science
- Ecological, evolutionary & environmental sciences
- Life sciences

Furthermore, please align your manuscript with our format requirements, which are summarized on the following checklist: <https://www.nature.com/documents/commsj-phys-style-formatting-checklist-article.pdf> Communications Earth & Environment formatting checklist

and also in our style and formatting guide <https://www.nature.com/documents/commsj-phys-style-formatting-guide-accept.pdf> Communications Earth & Environment formatting guide .

***** DATA:** Communications Earth & Environment endorses the principles of the Enabling FAIR data project (<http://www.copdess.org/enabling-fair-data-project/>). We ask authors to make the data that support their conclusions available in permanent, publically accessible data repositories. (Please contact the editor if you are unable to make your data available).

All Communications Earth & Environment manuscripts must include a section titled "Data Availability" at the end of the Methods section or main text (if no Methods). More information on this policy, is available at <http://www.nature.com/authors/policies/data/data-availability-statements-data-citations.pdf>

If a community resource is unavailable, data can be submitted to generalist repositories such as <https://figshare.com/> Figshare or <http://datadryad.org/> Dryad Digital Repository. Please provide a unique identifier for the data (for example a DOI or a permanent URL) in the data availability statement, if possible. If the repository does not provide identifiers, we encourage authors to supply the search terms that will return the data. For data that have been obtained from publically available sources, please provide a URL and the specific data product name in the data availability statement. Data with a DOI should be further cited in the methods reference section.

REVIEWER COMMENTS:

Reviewer #1 (Remarks to the Author):

This is an interesting manuscript about Campi Flegrei caldera in which the authors have applied advanced thermal and petrological models to evaluate the overpressure of the system and the amount of eruptible magma to provide plausible future scenarios of the ongoing unrest. I think that this manuscript is of high quality and of broad interest and impact and thus, it is suitable for Communications Earth & Environment.

I have only minor comments. Below I reported the main comments, while I put the others directly in the annotated manuscript (see attached file).

Lines-59-66: Do you mean that each intrusion caused an increase of T above the T reached by the previous intrusion, or that

each intrusion maintained the T above the pre-1950 T? This is something that it is not easy to understand from Figure 2. Maybe you can add in the supplementary a simpler figure showing the average T of the system over the time that you got from your thermal models (average T in the y-axis and time in the x-axis).

Figure 2: What is the dark blue curve? The total volume of magma of the shallow (4 km) reservoir? If yes, why peaks are not consistent with peaks of uplift? If the dark blue curve is only the volume of the excess fluids, maybe you can consider adding to the figure also the total volume (total V of magma or total V of magma + excess fluids). In addition, it would be worth adding in the main text (e.g., around lines 59-89) few more details about figure 2 to help the non-petrologist reader its comprehension. For example, I would briefly explain that, due to the combination of magma injection, cooling, latent heat of crystallization etc., the velocity of each geothermal curve is different and that's why the peaks of volumes at 1075 °C occur at time different to the peaks of volumes at 975 °C etc.

Line 93: The first time that I read the manuscript I was a bit confusing regarding lines 93-111, since I thought the authors assumed that all the overpressures were due to the exsolution of magmatic fluids only, neglecting the contribution from the new injected magma, even though their model assumes the new injection of magma (as written at line 37). Then, by reading lines 112-121, I understood the reason of why they have done this. However, I think that the authors should better introduce lines 93-111 to do not confuse the reader, by explicitly writing why they first calculated only the overpressures due to the exsolution of magmatic fluids (something like "To understand if the volume increases generated by the exsolution of magmatic fluids is enough to trigger an eruption and to explain geodetic data, we calculated the increase of volume and overpressures associated with the crystallisation-induced exsolution of magmatic fluids ...").

Equation 2: The Authors should specify that the equation 2 (written in this way) is valid for spherical magma bodies. Otherwise, equation 2 becomes $\tau = \gamma * (\eta_{\text{crust}}/G)$; where γ is a factor dependent on both the geometry of the magma body and the viscoelastic structure and is about equal to 1 for spherical magma bodies and ≥ 10 for sills (see Sigmundsson et al., 2020 <https://doi.org/10.1038/s41467-020-16054-6> or Galetto et al., 2022 <https://doi.org/10.1038/s41561-022-00960-z>). Actually, the authors could add a couple of lines to discuss how this assumption might affect their results.

In the Section "Mechanisms of reservoir's overpressurisation and temporal evolution of overpressure", I think that another point that the Authors should discuss is the fact that the overpressures estimated with their approach might be overestimated since they assumed constant mechanical properties of the host rock. 3D FEM models show a reduction in the estimated overpressures by assuming changes in the mechanical properties with temperature (e.g.; see further details in Gregg et al., 2018 <https://doi.org/10.1029/2018GL080393>; Zhan et al., 2019 <https://doi.org/10.1029/2019JB018681>).

The supplementary material files (text, figures and tables) are never quoted in the main text (and in the method section). I think that they should be mentioned somewhere.

I hope that these comments could further improve the quality of this manuscript.
Federico Galetto

Reviewer #2 (Remarks to the Author):

I've read this paper with great interest as it represents a very well-reasoned exercise about the "maximum" volume of magma that might be involved in the long-standing unrest at Campi Flegrei volcano (Southern Italy). I interpret this study as a contribution putting upper boundary conditions, in terms of size and effects (overpressure), about shallow magma that may have already emplaced at Campi Flegrei.

The Authors make clear since the beginning that they depict a sort of extreme scenario: if the three (main) events of ground uplift that have characterized the caldera floor displacement during the last 75 years were due to magma emplacement at shallow depth (about 4km), what would be the volume of magma involved and currently stored there? And what would make that cumulated volume eruptible?

The Authors do not want to address the complexity of Campi Flegrei unrest (they are neutral about it) but provide estimates of the magma left if the shallow magma scenario is accounted for. Given this assumption, not proven to be true for the current unrest, the work is presented by developing calculations in a logical and consistent way, although I see some limits that I am going to put in evidence below and which can be fixed with little additional effort.

The main implication of this study rests in the following sentence, extracted from the abstract: "*We show that the ascent of magma and magmatic fluids from depth is necessary to explain the measurements collected since 1950*". This is indeed the main point, as this statement refers to a conclusive argument which does not need any matching/fitting of the ground displacement or geochemical patterns. It comes logically from the study itself, with its premises and has an epistemic value that should not be underrated.

As for current unrest, the Authors are not taking a definite position (not needed) but under the a priori assumption made, they evaluate timing and conditions to reach an eruptive threshold. The entire study then looks like the evaluation of a "worst case scenario at 4 km depth", without neglecting the arguments for the suppression of internally triggered eruptions. These arguments, thus, may apply in future (note that the Authors evaluate a time span of two decades) and have applied to past unrest episodes. Let me stress again that there is nothing here about a future eruption and its style, rather, there are

boundaries to put at 4 km depth about the conditions which may lead to an eruption.

This said, I have of course some remarks

-Line 41-43: The Authors could state that they provide a computation about the physical state of the maximum amount of magma potentially stored at 4 km depth: It has to be a maximum one simply because when you neglect the poro-elastic response of the hydrothermal system and ascribe the entire subsidence to the loss of magmatic volatiles, you are clearly scaling the entire deflation to the maximum loss of magmatic volatiles. This maximum loss is associated with a maximum amount of emplaced magma at shallow depth. That's why this paper is important: assuming the magmatic scenario alone (the "worst scenario at 4 km depth"), it leads to the maximum amount of magma by using basic properties about volatile-rich magmas and their thermal (time) evolution.

Such a maximum amount of magma then depends on adopted numbers and how these are parameterized. This leads to me to point out that a more detailed sensitivity study should be included.

- Line 44-46: That's why it is a maximum value. Actually, rather than a cross-check, the 1985-2004 deflation is part of the Authors' computation. The cross-check in my view should be made against something different such as density. So here is a major point: why do not they use density? To my knowledge, two out of five authors are experts of gravimetry and run gravity monitoring. They must have some density values to cross-check or even validate their approach here. This addition would be a great added-value for the paper.

- Line 50 on: Moretti et al. (2013 EPSL, vol. 367, 95-104) simulated the emplacement of 107 m³ (2x10¹¹ Kg/m³) from gravimetry of magma at 4 km. They used a bulk density of 2008 kg/m³ (due to the excess fluid), total (exsolved + dissolved) volatile contents of H₂O (3.5 ± 0.35 wt%), CO₂ (2 ± 0.46 wt%), S (0.15 ± 0.21 wt%). Gas mass (initial, after emplacement) was 3 wt%, volume fraction of gas was 0.21. They included crystallization over a short time window and much more. They compared with fluxes (the Authors mention this aspect) rather than the volume compatible with deflation simply because CO₂ flux is a routine measurement. One of their outcomes was that any magma emplacement compatible with physical observations at surface would have not been enough to explain the amount of discharged fluids and that a deep contribution was required. That deep contribution was modelled, btw. I think that in that paper there are numbers useful for comparison and surely arguments for discussion of previous models in which a volatile-rich magma emplaced at 4 km was used to explain data. You should also, in my opinion, consider the results from Aiuppa et al. (2013, G3) and also Caliro et al. (2014 GCA). These suggestions are not only aimed at giving credit to previous studies, but to offer the possibility for a larger discussion in which results from this paper emerge better.

Also note that it is plenty of studied on melt inclusions in which total volatiles have been assessed, such as Arienzo et al. (2010 CHEMGE) and many others that I will not list here, but that it is worth considering to constrain the initial amount and composition of so-called "excess" fluid, hence compressibility.

In line with this, I wonder about the effect of CO₂, which at CFC is central in driving fluid saturation in magmas, but which is not considered in this study. Therefore, in terms of "excess" fluid and compressibility I expect huge variations, which here are not discussed

- The justification for a viscosity around 10¹⁷ is understandable but physically vague. Furthermore, G too has significant variations, reflecting rock damage: However, the η/G ratio is a central parameter to estimate overpressure. Other possibilities might be related to "hardening" effects on G (some literature, Vanorio group in Stanford, has proposed this explanation). Perhaps a combination of effects is the best explanation. Still, 0.26 years to me looks too small, but I must recognize that this is just a feeling. Anyway, I suggest to provide more arguments and possibly a parametric assessment (upper and lower thresholds, best value), which I think it should be visualized in fig 4.

The Authors certainly understand the importance of such a request, given the fact that they insist on the potential for internally triggered eruptions and state that "*Our calculations suggest that potentially eruptible magma is present today at ~4 km depth, but that eruption may be suppressed by the combination of small reservoir volume, magma compressibility, and viscous deformation of the surrounding crust*".

Other comments:

-About the title. In my opinion "Physics-informed" is not appropriate although appealing. The current title generates expectations about scientific (machine) learning processes encoding model equations as components of a neural network. Indeed, from the title my expectation was to read about a learning tool harnessing observational data to forecast the unrest evolution. Above, I have used the word "epistemic". I think it should appear in the title as related to the boundaries to adopt in the eventuality of magma involvement.

- The last sentence sounds a little bit obvious, and not worth of being the conclusion of this paper. The same for the last sentence of the abstract. Anything better or more specific?

- Line 98. Please use only delta notation (ΔV instead of dV). It does not seem to me that integration of a differential equation follows: Also the Y-axis of fig. 3a should be labelled $\Delta V/\Delta t$: Same notation for eq. 3 at line 191,

- Campi Flegrei is certainly one of the most studied volcanoes of the world. It is of course impossible to give proper credits to all previous work. There is however the need to cite some papers, but also to use their scientific content. Below some remarks/suggestions which can be useful for this study

1) Lines 27-30: "*The mechanism generating inflation is still debated... omissis... plumbing system*". It should be made clear

that what stated in text by Authors applies only to the current unrest, otherwise misconceptions and reasoning shortcuts keep propagating when they should not. In particular, the interpretation proposed in ref. 7 (plus other refs. below), based on geochemical data available until 2016, has its founding argument on the contrasting geochemical behaviors shown by the 1982-84 and post-2005 crises.

Note that in ref. 7, as well as in Moretti et al., 2013 EPSL, Moretti et al. 2018 (Sci. Reports) and Troise et al. (2019, ESR) the previous 1982-84 crisis is interpreted as strongly driven by the emplacement of a volatile-rich magma at 4 km depth, which leaves its residual of about 1 m at the end of the 1985-2004 subsidence (Moretti et al., 2018 Sci. Rep.; Troise et al., 2019 ESR).

The contrasting geochemical behavior shown by the 1982-84 unrest vs the current one cannot be overlooked, because it is intrinsic to the data, is not derived from models or elaborations. It has profound implications, let's say an epistemic meaning, and despite being neglected by the literature cannot however be avoided. For more info please go through Moretti et al. (2020; Dev. in Volcanology book; Belkin, De Vivo, Rolandi eds.), who expand the reasoning introduced by Caliro et al. (2014, GCA).

2) Lines 33-34: Bevilacqua and coworkers have done the same and given values (Bevilacqua, A., Neri, A., De Martino, P., Isaia, R., Novellino, A., Tramparulo, F. D. A., & Vitale, S., 2020. Journal of Geodesy, 94, 1-27.). Please use it, at least to check

Roberto Moretti

Reviewer #3 (Remarks to the Author):

Summary:

This paper presents a model of evolving pressure, temperature, and gas contents in the shallow Campi Flegrei magma reservoir since 1950, to examine whether sufficient overpressure to induce eruptions is currently present or existed over the past decades of inflationary events.

Quantifying conditions in the Campi Flegrei magma system is an important goal. However, I am unsure how one of the key figures in the paper (Fig. 2) is intended to be interpreted. Additionally, some of the model result appear unintuitive, for example the observation that this model seems to allow continued intrusions of material with no net increases in overpressure, but with no stated mechanisms for viscous relaxation or fluid loss incorporated in the model. The authors provided minimal information about the model, either from the main text or the supplement. I am thus unsure how to evaluate the model or understand the results. If this work is extended from previous models, then references should be clearly provided. If not, then much more detail is needed in this publication for readers to be able to interpret the results presented, given the potential societal significance of the claims the authors make about the current state of the volcano.

Detailed comments:

L10: comma not needed

Fig 2: It would be helpful here to add a subplot showing some of the data used to derive these model results, e.g., a timeseries of deformation data or inverted volume changes, or to add an earlier figure showing such data. Otherwise readers will have no sense of what the actual data looks like without going to other publications.

Fig 2: This figure is very difficult to interpret. It seems like the blue color is meant to map to the right (blue) axis, and the other colors map to the left axis. However, if that is the case, it is unclear why magma volume would decrease to zero very rapidly following injections of hotter magma but remain elevated for much longer following injections of cooler magma. The caption and/or figure needs to be modified to make clear exactly what is being plotted and what corresponds to what axes. If the blue lines that map to the left (black) axis are meant to represent magma volumes, it is then unclear how this relates to the other three excess fluid curves, if there are three distinct scenarios (with different injection temperatures) being shown. The supplement indirectly implies that these curves might instead represent the volume of magma within certain temperature bins, but nothing in the figure or caption clarifies this.

Fig 3a: for the first injection, is it correct that the volume increase from second boiling is larger than the volume increase from the initial injection?

Fig 3b: why does compressibility appear to have an upper threshold at $\sim 3E-9$?

L111: Changing bulk magma compressibility will complicate things, but in general it seems that once a fluid phase is present the compressibility effects would not be able to indefinitely counter the effects of continued magma injection, and so pressure on the surrounding crust should begin increasing on average as magma input continues, since the authors state the model does not include fluid loss or viscous relaxation. This might relate to how the authors are treating solidified magma and its contributions to the total volume of crustal intrusion, but I am not clear on this point from the information provided.

L142: This seems like an unnecessary level of approximation, it would be more meaningful, and seemingly relatively straightforward, to explicitly include viscous relaxation in the model.

L164: These seem like unrealistically small bounds on reservoir volume.

L201: Fixed temperature boundary conditions would be most reasonable in the presence of relatively efficient hydrothermal circulation, is that the assumption here?

L203: This statement seems to indicate that only one injection temperature is assumed, but how does this relate to the different temperatures in Fig 2?

L207: Why these two choices, which are both relatively thin? It seems like a thicker sill would provide a better end-member estimate.

Reviewer #4 (Remarks to the Author):

Thank you for the opportunity to read this comprehensive study on the evolution of unrest at Campi Flegrei caldera and its implications for future eruption scenarios. The paper addresses an important topic with significant implications for understanding volcanic systems and mitigating risks to the densely populated region surrounding the caldera. While the study provides valuable insights, there are several aspects that require clarification, additional validation, and further elaboration to strengthen the robustness and impact of your findings.

The main conclusion of the paper is that while potentially eruptible magma has accumulated at a depth of approximately 4 km beneath Campi Flegrei caldera (Italy), the likelihood of an eruption being triggered solely by internal processes within the magma reservoir is currently low. This is due to several mitigating factors:

1. Small reservoir volume: The shallow magma reservoir is relatively small, which reduces the capacity for sufficient overpressure to trigger an eruption.
2. High magma compressibility: The presence of excess fluids increases magma compressibility, which dampens overpressure within the reservoir.
3. Viscous deformation of the crust: The surrounding crust's high temperature and low viscosity allow for deformation that accommodates reservoir expansion without generating critical overpressure.

The study indicates that continued magma injection at the current rate ($8 \times 10^6 \text{ m}^3/\text{year}$) could lead to conditions favorable for an eruption within the next two decades. Additionally, the long-term integrity of the overlying crust is a critical factor. Fractures caused by decades of unrest, deformation, and hydrothermal activity could eventually form pathways for magma to reach the surface, potentially triggering an eruption even if the internal overpressure and reservoir size are suboptimal.

My main concern is about the uncertainty of the assumptions, for example:

The paper assumes a simplified geometry for the magma reservoir (e.g., thin sills at a fixed depth of 4 km). However, natural magmatic systems are often complex, with interconnected chambers and variable depths. How sensitive are the results to changes in the assumed geometry or depth of the reservoir? What is the uncertainty regarding the statement: "We assume a proportionality between the total measured uplift and the corresponding increase in volume estimated for the last two episodes (Table 1)." Especially, how to justify the diameters of the sills?

The thermal and mechanical models assume the reservoir behaves as a closed system for fluid release and pressure calculations. How realistic is this assumption, given that Campi Flegrei has a well-developed hydrothermal system that likely allows for fluid escape? How would incorporating open-system behavior affect the conclusions? In Line 86: I don't quite understand:

"However, even if half of the excess fluids would have been lost in the period 1950–1983, the excess fluid volume in the shallow reservoir would still be sufficient to explain the deflation episode (Fig. 2)." If the excess fluids were degassing during 1950–1983, should we observe deflation during that time, or was the inflation due to magma injection partially compensated by degassing? Also, can the degassing of the excess fluids explain the deflation observed during 1984–2005?

Why is it valid to use the rock properties at 3 km depth (Line 128) to represent the stiffness of the entire volcanic system? I believe using a numerical model that couples thermal and mechanical processes would be more convincing for explaining the relationship between deformation and overpressure evolution.

Another concern is about the method part; it is hard to reproduce the model result due to lack of details. For example:

1. Which numerical method is used: finite volume or finite element? which code?
2. What is the geometry of the model?
3. What is the initial geothermal gradient ($135^\circ\text{C}/\text{km}$)? While this gradient may be applicable to the rock above the magma reservoir, is it valid for the far field?

Some minor comments:

Figure 2: Need a more detailed caption to describe which y-axes those curves belong to. Color-coding the shadows of different temperature cases with a sequential colormap to indicate the temperature from low to high would make the figure more readable.

Line 27: I suggest adding a time series plot of deformation to Figure 2.

Please check the format of the values and their units. In some cases, there is a space between the value and the unit, while in others, there is none. Additionally, the unit is missing in at least one instance (Line 53).

benefits**

Communications Earth & Environment is committed to improving transparency in authorship. As part of our efforts in this direction, we are now requesting that all authors identified as 'corresponding author' create and link their Open Researcher and Contributor Identifier (ORCID) with their account on the Manuscript Tracking System prior to acceptance. ORCID helps the scientific community achieve unambiguous attribution of all scholarly contributions. You can create and link your ORCID from the home page of the Manuscript Tracking System by clicking on 'Modify my Springer Nature account' and following the instructions in the link below. Please also inform all co-authors that they can add their ORCIDs to their accounts and that they must do so prior to acceptance.

Version 1:

Decision Letter:

Dear Professor Caricchi,

Please allow us to apologise for the delay in reaching a decision on your manuscript titled "Scenario-based forecast of the evolution of 75 years of unrest at Campi Flegrei caldera (Italy)". It has now been seen again by 4 reviewers, and we include their comments at the end of this message. They find your work of interest, but Reviewer #3 in particular retains some important concerns. We remain interested in the possibility of publishing your study in Communications Earth & Environment, but would like to consider your responses to these concerns and assess a revised manuscript before we make a final decision on publication.

Specifically, in the revised manuscript please ensure that your modelling approach is explained in sufficient detail so as to allow the work to be reproducible and the validity of the results to be effectively assessed.

We therefore invite you to revise and resubmit your manuscript, along with a point-by-point response that takes into account the points raised. Please highlight all changes in the manuscript text file.

Please submit your point-by-point responses as a separate file, distinct from your cover letter where you can add responses to the Editors' comments that you do not want to be made available to the reviewers. Word files are preferred. We recommend that any figures, tables or graphs that are included in the response to reviewers are also included in the main article or Supplementary Information.

Please use the following link to submit your revised manuscript, point-by-point response to the referees' comments (which should be in a separate document to any cover letter), a tracked-changes version of the manuscript (as a PDF file) and the completed checklist:

Link Redacted

We hope to receive your revised paper within six weeks; please let us know if you aren't able to submit it within this time so that we can discuss how best to proceed. If we don't hear from you, and the revision process takes significantly longer, we may close your file. In this event, we will still be happy to reconsider your paper at a later date, as long as nothing similar has been accepted for publication at Communications Earth & Environment or published elsewhere in the meantime.

Please do not hesitate to contact us if you have any questions or would like to discuss these revisions further. We look forward to seeing the revised manuscript and thank you for the opportunity to review your work.

Best regards,

Joe Aslin

Deputy Editor,
Communications Earth & Environment

Consulting Editor,

EDITORIAL POLICIES AND FORMATTING

- Behavioural and social science
- Ecological, evolutionary & environmental sciences
- Life sciences

Furthermore, please align your manuscript with our format requirements, which are summarized on the following checklist: <https://www.nature.com/documents/commsj-phys-style-formatting-checklist-article.pdf> Communications Earth & Environment formatting checklist

and also in our style and formatting guide <https://www.nature.com/documents/commsj-phys-style-formatting-guide-accept.pdf> Communications Earth & Environment formatting guide .

***** DATA:** Communications Earth & Environment endorses the principles of the Enabling FAIR data project (<http://www.copdess.org/enabling-fair-data-project/>). We ask authors to make the data that support their conclusions available in permanent, publically accessible data repositories. (Please contact the editor if you are unable to make your data available).

All Communications Earth & Environment manuscripts must include a section titled "Data Availability" at the end of the Methods section or main text (if no Methods). More information on this policy, is available at <http://www.nature.com/authors/policies/data/data-availability-statements-data-citations.pdf>.

If a community resource is unavailable, data can be submitted to generalist repositories such as <https://figshare.com/> or <http://datadryad.org/> Dryad Digital Repository. Please provide a unique identifier for the data (for example a DOI or a permanent URL) in the data availability statement, if possible. If the repository does not provide identifiers, we encourage authors to supply the search terms that will return the data. For data that have been obtained from publically available sources, please provide a URL and the specific data product name in the data availability statement. Data with a DOI should be further cited in the methods reference section.

REVIEWER COMMENTS:

Reviewer #1 (Remarks to the Author):

I think that the Authors have properly addressed all the comments in the revised manuscript. I have no further comments. The only thing is that the Authors should check the supplementary figure S1, since there is one curve that goes outside the figure, and I am not sure if it is correct or if it is a plotting error.

Federico Galetto.

Reviewer #2 (Remarks to the Author):

The Authors have well responded and I compliment them for this.

Although I understand their point, I still think that a mention to density should be included. It is in fact quite logical to expect a cross-check against density, so this probably deserves at least a mention, even of why it was not considered. The argument in their rebuttal could be used for this purpose. However, the Authors should feel free to accept or decline my suggestion.

The mention of the role played by CO₂ is enough for the purposes of this paper, despite inclusion of this species certainly shifts Author's calculations.

Reviewer #3 (Remarks to the Author):

In my initial review, I asked how the authors' model accommodates continued magma intrusions with no net pressure increase but no means of releasing pressure via fluid loss or viscous relaxation. I do not think the authors have addressed this concern or provided sufficient information for readers to reproduce or understand their model. Given that the model is the main focus of the paper, I am still unable to effectively evaluate the paper.

In response to my question, the authors state that their model considers instantaneous rather than continuous magma intrusion. However, added material needs to be accommodated by the crust whether it is intruded continuously or in discrete events. The authors next state that they modified a sentence to no longer mention overpressure, which does not address my fundamental question about their model.

Very generally, if material is intruded into some location in the crust, the crust must deform to accommodate the added material. If there is no inelastic crustal deformation, then this crustal deformation should result in elastic stress changes and increased pressure on the intruded material. The distribution of stress in the surrounding crust and the solidifying intrusion might change in complicated ways during solidification, but stress changes should still be present in the solidified intrusion and crust, and should contribute to pressure in any remaining fluid and/or subsequent intrusions. The authors state that in their model "the material below each sill is advected downward" to make room for each sill, presumably invoking some sort of "artificial inelastic deformation event" since no additional details are provided. Even if this approach is justifiable in some limits, e.g., when sufficient time has passed for most stresses from the previous intrusion to have relaxed away inelastically, the authors do not provide such justifications, and this would still not explain how fluid overpressure is able to drop to zero between intrusions. The only equation provided for overpressure in the authors' model is the simple relation $dp = dV/(\beta \cdot V)$. In this equation, dp can only approach zero if either dV approaches zero, which is not consistent with the authors' figures that show a finite volume of fluid persisting (even if the authors were to effectively justify neglecting solidified material), or if β or V approach infinity. The authors state that they treat the "the size of the magma reservoir as the calculated volume of magma above its solidus temperature", which does not appear to approach infinity from the provided plots. The authors also provide plots of β , showing maximum values of $\sim 3E-9$. So regardless of what the physical behavior of the system should be, it is unclear how the authors' model produces the behavior they show based on the limited number of equations or other model descriptions provided.

Reviewer #4 (Remarks to the Author):

Thank you for addressing my comments and concerns. I am happy to see that the authors have addressed all of my questions with thorough answers and pertinent revisions after closely examining the updated manuscript. There is now more clarity in the methodology, figures, and overall scientific interpretation of the manuscript. I am satisfied with the changes and now recommend the publication of the manuscript in its current form.

** Visit Nature Portfolio's author and referees' website at www.nature.com/authors for information about policies, services and author benefits**

Communications Earth & Environment is committed to improving transparency in authorship. As part of our efforts in this direction, we are now requesting that all authors identified as 'corresponding author' create and link their Open Researcher and Contributor Identifier (ORCID) with their account on the Manuscript Tracking System prior to acceptance. ORCID helps the scientific community achieve unambiguous attribution of all scholarly contributions. You can create and link your ORCID from the home page of the Manuscript Tracking System by clicking on 'Modify my Springer Nature account' and following the instructions in the link below. Please also inform all co-authors that they can add their ORCIDs to their accounts and that they must do so prior to acceptance.

Version 2:

Decision Letter:

Dear Professor Caricchi,

Your manuscript titled "Scenario-based forecast of the evolution of 75 years of unrest at Campi Flegrei caldera (Italy)" has now been seen by our reviewer, whose comments appear below. In light of their advice we are delighted to say that we are happy, in principle, to publish a suitably revised version in Communications Earth & Environment.

We therefore invite you to revise your paper one last time to address the remaining concerns of our reviewer and incorporate realistic uncertainty assessments. At the same time we ask that you edit your manuscript to comply with our format requirements and to maximise the accessibility and therefore the impact of your work.

EDITORIAL REQUESTS:

*****Please take care to match our formatting and policy requirements. We will check revised manuscript and return manuscripts that do not comply. Such requests will lead to delays. *****

SUBMISSION INFORMATION:

OPEN ACCESS:

Communications Earth & Environment is a fully open access journal. Articles are made freely accessible on publication. For further information about article processing charges, open access funding, and advice and support from Nature Portfolio, please visit <https://www.nature.com/commsenv/open-access>

Link Redacted

Best regards,

Joe Aslin

Deputy Editor,
Communications Earth & Environment

Consulting Editor,
Communications Sustainability

<https://www.nature.com/commsenv/>
Twitter: @CommsEarth

REVIEWERS' COMMENTS:

Reviewer #3 (Remarks to the Author):

The revised manuscript is satisfactory. The authors have removed plots and discussions of their overpressure calculations that were potentially problematic.

The authors could consider adding error bars to their estimates of timescale ratios (i.e., the red star in their new figure 3). Currently the authors just state in the caption that this result is insensitive to variation in parameters, but realistic assessments of the uncertainty in various parameters involved, e.g., "effective crustal viscosity" or magma reservoir shape (which the authors do briefly mention elsewhere in the text) would presumably shift some timescales by an order of magnitude or more. Showing even extremely simple upper and lower bound estimates (with brief explanations in the text somewhere) would help indicate how robust the conclusions are.

** Visit Nature Portfolio's author and referees' website at www.nature.com/authors for information about policies, services and author benefits**

Dear Editor,

You can find in the following our point-to-point answer to the Reviewers' queries. Their comments were extremely useful and improved the quality of our manuscript.

Our replies are in *italic* and line numbering of the replies refers to the track-changed version of the manuscript.

Reviewer #1 (Remarks to the Author):

This is an interesting manuscript about Campi Flegrei caldera in which the authors have applied advanced thermal and petrological models to evaluate the overpressure of the system and the amount of eruptible magma to provide plausible future scenarios of the ongoing unrest. I think that this manuscript is of high quality and of broad interest and impact and thus, it is suitable for Communications Earth & Environment. I have only minor comments. Below I reported the main comments, while I put the others directly in the annotated manuscript (see attached file).

Lines-59-66: Do you mean that each intrusion caused an increase of T above the T reached by the previous intrusion, or that each intrusion maintained the T above the pre-1950 T? This is something that it is not easy to understand from Figure 2. Maybe you can add in the supplementary a simpler figure showing the average T of the system over the time that you got from your thermal models (average T in the y-axis and time in the x-axis).

We meant that the time interval was not sufficient for the region where the magma injection occurs, to cool down to the pre 1950 temperature. We modified the text to clarify (Lines 94-95). We did not add the average temperature as Figure 2 shows the volume of magma at 875°C, which is above the solidus. We specify now in the text so that the reader can follow better.

Figure 2: What is it the dark blue curve? The total volume of magma of the shallow (4 km) reservoir? If yes, why peaks are not consistent with peaks of uplift? If the dark blue curve is only the volume of the excess fluids, maybe you can consider adding to the figure also the total volume (total V of magma or total V of magma + excess fluids).

The dark blue curve shows the total volume of excess fluids. We modified the figure adding a legend and also explain better in the figure caption. We would need to adjust the left y-axis, which would compress significantly the figure not allowing to clearly see the evolution of the volumes of magma at different temperatures. Nevertheless, we added the volume of magma injected at each episode of magma input and added this to the caption accordingly.

In addition, it would be worth adding in the main text (e.g., around lines 59-89) few more details about figure 2 to help the non-petrologist reader its comprehension. For example, I would briefly explain that, due to the combination of magma injection, cooling, latent heat of crystallization etc., the velocity of each geothermal curve is

different and that's why the peaks of volumes at 1075 °C occur at time different to the peaks of volumes at 975 °C etc.

Excellent point. We added the text as suggested (Lines 95-97).

Line 93: The first time that I read the manuscript I was a bit confusing regarding lines 93-111, since I thought the authors assumed that all the overpressures were due to the exsolution of magmatic fluids only, neglecting the contribution from the new injected magma, even though their model assumes the new injection of magma (as written at line 37). Then, by reading lines 112-121, I understood the reason of why they have done this. However, I think that the authors should better introduce lines 93-111 to do not confuse the reader, by explicitly writing why they first calculated only the overpressures due to the exsolution of magmatic fluids (something like “To understand if the volume increases generated by the exsolution of magmatic fluids is enough to trigger an eruption and to explain geodetic data, we calculated the increase of volume and overpressures associated with the crystallisation-induced exsolution of magmatic fluids ...”).

Thanks for the comment. We modified the text following the suggestion (Lines 157-158).

Equation 2: The Authors should specify that the equation 2 (written in this way) is valid for spherical magma bodies. Otherwise, equation 2 becomes $\tau = \gamma * (\eta_{\text{crust}}/G)$; where γ is a factor dependent on both the geometry of the magma body and the viscoelastic structure and is about equal to 1 for spherical magma bodies and ≥ 10 for sills (see Sigmundsson et al., 2020 <https://doi.org/10.1038/s41467-020-16054-6> or Galetto et al., 2022 <https://doi.org/10.1038/s41561-022-00960-z>). Actually, the authors could add a couple of lines to discuss how this assumption might affect their results.

We would like to thank the reviewer for pointing this out. We added the suggested references to the main text and added the calculations for non-spherical source Lines 225, 251-258).

In the Section “Mechanisms of reservoir’s overpressurisation and temporal evolution of overpressure”, I think that another point that the Authors should discuss is the fact that the overpressures estimated with their approach might be overestimated since they assumed constant mechanical properties of the host rock. 3D FEM models show a reduction in the estimated overpressures by assuming changes in the mechanical properties with temperature (e.g.; see further details in Gregg et al., 2018

<https://doi.org/10.1029/2018GL080393>; Zhan et al., 2019

<https://doi.org/10.1029/2019JB018681>).

We added a sentence about this and one of the suggested references (Lines 270-272).

The supplementary material files (text, figures and tables) are never quoted in the main text (and in the method section). I think that they should be mentioned somewhere.

Thanks for this, we now refer to the supplementary material in the main text (Methods section, Lines 338, 340-343).

Comments by Reviewer 1 in the annotated pdf of the manuscript not already addressed in the general comments

I would better emphasize the fact the 2005-present uplift not only have recovered all the subsidence occurred from 1984 to 2005, but in the last years the uplift has exceeded the peak of 1984 and thus it is at its maximum height (at least since the beginning of the unrest in 1950).

Added a sentence (Lines 28-36).

Do you have data supporting the fact that the deformation is associated to the same source at the same/similar depth? (e.g., the levelling measurements show the same deformation pattern).

No and we now specify this in the text (Line 44).

add the range (of critical overpressure)?

Done (Line 269).

M>5.0 earthquakes can trigger an eruption (see Gregg et al., 2018

<https://doi.org/10.1029/2018GL080393>; Gregg et al., 2022)

While we think that an earthquake could initiate dyke propagation, we think that the main limiting factor would still be size of the reservoir (too small to allow the dykes to reach the surface before the pressure drops). For this reason we did not added any text related to this comment.

Reviewer 2

I've read this paper with great interest as it represents a very well-reasoned exercise about the "maximum" volume of magma that might be involved in the long-standing unrest at Campi Flegrei volcano (Southern Italy). I interpret this study as a contribution putting upper boundary conditions, in terms of size and effects (overpressure), about shallow magma that may have already emplaced at Campi Flegrei. The Authors make clear since the beginning that they depict a sort of extreme scenario: if the three (main) events of ground uplift that have characterized the caldera floor displacement during the last 75 years were due to magma emplacement at shallow depth (about 4km), what would be the volume of magma involved and currently stored there? And what would make that cumulated volume eruptible? The Authors do not want to address the complexity of Campi Flegrei unrest (they are neutral about it) but provide estimates of the magma left if the shallow magma scenario is accounted for. Given this assumption, not proven to be true for the current unrest, the work is presented by developing calculations in a logical and consistent way, although I see some limits that I am going to put in evidence below and which can be fixed with little additional effort. The main implication of this study rests in the following sentence, extracted from the abstract: "We show that the ascent of magma and magmatic fluids from depth is necessary to explain the measurements collected since 1950". This is indeed the main point, as this statement refers to a conclusive argument which does not need any matching/fitting of the ground displacement or geochemical patterns. It comes logically from the study itself, with its premises and

has an epistemic value that should not be underrated. As for current unrest, the Authors are not taking a definite position (not needed) but under the a priori assumption made, they evaluate timing and conditions to reach an eruptive threshold. The entire study then looks like the evaluation of a “worst case scenario at 4 km depth”, without neglecting the arguments for the suppression of internally triggered eruptions. These arguments, thus, may apply in future (note that the Authors evaluate a time span of two decades) and have applied to past unrest episodes. Let me stress again that there is nothing here about a future eruption and its style, rather, there are boundaries to put at 4 km depth about the conditions which may lead to an eruption.

The reviewer has perfectly captured the philosophy of our contribution to the ongoing discussion on the unrest at Campi Flegrei.

-Line 41-43: The Authors could state that they provide a computation about the physical state of the maximum amount of magma potentially stored at 4 km depth: It has to be a maximum one simply because when you neglect the poro-elastic response of the hydrothermal system and ascribe the entire subsidence to the loss of magmatic volatiles, you are clearly scaling the entire deflation to the maximum loss of magmatic volatiles. This maximum loss is associated with a maximum amount of emplaced magma at shallow depth. That’s why this paper is important: assuming the magmatic scenario alone (the “worst scenario at 4 km depth”), it leads to the maximum amount of magma by using basic properties about volatile-rich magmas and their thermal (time) evolution.

Such a maximum amount of magma then depends on adopted numbers and how these are parameterized. This leads to me to point out that a more detailed sensitivity study should be included.

We agree with the reviewer. We added a sentence about this (Lines 56-59). A part of the results of the sensitivity study was added in Figure 3, where we consider a sill of 15 m thickness (and larger diameter) instead of a sill of 25 m thickness. For a thickness of 15m no eruptible magma would be present today and as this is an extremely important result, we added a supplementary figure and commented on the main text about the impact of decreasing the sill thickness from 25 to 15m (Lines 87-92).

We also performed additional simulation for thicknesses of 100 and 200 meters, which however are significantly departing from any constrain obtained from geodetic and geophysical methods and therefore we did not add the results for these simulations (which show significantly larger volumes of eruptible magma would be present today at 4 km depth).

- Line 44-46: That’s why it is a maximum value. Actually, rather than a cross-check, the 1985-2004 deflation is part of the Authors’ computation. The cross-check in my view should be made against something different such as density. So here is a major point: why do not they use density? To my knowledge, two out of five authors are experts of gravimetry and run gravity monitoring. They must have some density values to cross-check or even validate their approach here. This addition would be a great added-value for the paper.

The sentence to which the Reviewer refers to has been modified. The observation of the review regarding gravity is correct, however, based on the gravity data collected during the subsidence phase following the 1984, it is difficult to obtain a reliable assessment of the density of redistributed mass in the crust. For example, the dg/dh data collected during the 1984–2011 period is highly scattered and affected by significant uncertainty. Although they generally indicate a decrease in density, an exception is observed at the Solfatara station, where the density increases and the mass redistribution appears unchanged (Gottsmann et al., 2003). For this reason, even a tentative evaluation of the mass density can result in significant errors. Thus, we decided not to modify the main text to integrate gravity.

- Line 50 on: Moretti et al. (2013 EPSL, vol. 367, 95-104) simulated the emplacement of 107m³ (2x10¹¹ Kg/m³) from gravimetry of magma at 4 km. They used a bulk density of 2008kg/m³ due to the excess fluid), total (exsolved + dissolved) volatile contents of H₂O (3,5 ± 0.35 wt%), CO₂ (2 ± 0.46 wt%), S (0.15 ± 0.21 wt%). Gas mass (initial, after emplacement) was 3 wt%, volume fraction of gas was 0.21. They included crystallization over a short time window and much more. They compared with fluxes (the Authors mention this aspect) rather than the volume compatible with deflation simply because CO₂ flux is a routine measurement. One of their outcomes was that any magma emplacement compatible with physical observations at surface would have not been enough to explain the amount of discharged fluids and that a deep contribution was required. That deep contribution was modelled, btw. I think that in that paper there are numbers useful for comparison and surely arguments for discussion of previous models in which a volatile-rich magma emplaced at 4 km was used to explain data. You should also, in my opinion, consider the results from Aiuppa et al. (2013, G3) and also Caliro et al. (2014 GCA). These suggestions are not only aimed at giving credit to previous studies, but to offer the possibility for a larger discussion in which results from this paper emerge better. Also note that it is plenty of studied on melt inclusions in which total volatiles have been assessed, such as Arienzo et al. (2010 CHEMGE) and many others that I will not list here, but that it is worth considering to constrain the initial amount and composition of so-called “excess” fluid, hence compressibility. In line with this, I wonder about the effect of CO₂, which at CFC is central in driving fluid saturation in magmas, but which is not considered in this study. Therefore, in terms of “excess” fluid and compressibility I expect huge variations, which here are not discussed

Another extremely important comment. We concur that the gas emissions are best explained if a deeper supply of magmatic fluids is included. Importantly, while the supply of fluids from the deeper reservoir is necessary to account for the gas emission data, it is not sufficient to explain the geodetic data as showed in Astort et al. (2024). We have now modified the text to include the discussion the reviewer hints to (Lines 132-147). The point about CO₂ is also very important. The flushing of CO₂-rich fluids from depth will result in two main effects: 1. The release of H₂O-rich fluids and 2. Magma crystallisation. We added few sentences about this (Lines 132-147). We have added also the references suggested by the reviewer as they are necessary to discuss the points suggested by the reviewer.

The justification for a viscosity around 10¹⁷ is understandable but physically vague.

Furthermore, G too has significant variations, reflecting rock damage: However, the η/G ratio is a central parameter to estimate overpressure. Other possibilities might be related to “hardening” effects on G (some literature, Vanorio group in Stanford, has proposed this explanation). Perhaps a combination of effects is the best explanation. Still, 0.26 years to me looks too small, but I must recognize that this is just a feeling. Anyway, I suggest to provide more arguments and possibly a parametric assessment (upper and lower thresholds, best value), which I think it should be visualized in fig 4. The Authors certainly understand the importance of such a request, given the fact that they insist on the potential for internally triggered eruptions and state that “Our calculations suggest that potentially eruptible magma is present today at ~ 4 km depth, but that eruption may be suppressed by the combination of small reservoir volume, magma compressibility, and viscous deformation of the surrounding crust”.

This is indeed a problematic quantity to define for the arguments exposed by the reviewer. Reviewer #1 correctly pointed out also that the relaxation timescale for non-spherical intrusion, is actually about 1 order of magnitude larger. For this reason in the main text we discuss the impact of a relaxation timescales between 0.26 and 2.6 years on our calculations (Lines 230-258). As both viscosity and shear modulus are difficult to constrain, we think this one order of magnitude range is sufficient to fully present our arguments.

Other comments:

About the title. In my opinion “Physics-informed” is not appropriate although appealing. The current title generates expectations about scientific (machine) learning processes encoding model equations as components of a neural network. Indeed, from the title my expectation was to read about a learning tool harnessing observational data to forecast the unrest evolution. Above, I have used the word “epistemic”. I think it should appear in the title as related to the boundaries to adopt in the eventuality of magma involvement.

If the editor concurs, the title could be change to: “Scenario-based forecast of the evolution of 75 years of unrest at Campi Flegrei caldera (Italy)” to remove the “physics-informed” as suggested by the reviewer.

The last sentence sounds a little bit obvious, and not worth of being the conclusion of this paper. The same for the last sentence of the abstract. Anything better or more specific?

We changed the last sentence of the abstract to point out another important results of our study (i.e. decades of magma input might be required to reach a reservoir size compatible with eruption and similar to the 1538 eruption of Monte Nuovo; Lines 18-21).

Line 98. Please use only delta notation (ΔV instead of dV). It does not seem to me that integration of a differential equation follows: Also the Y-axis of fig. 3a should be labelled $\Delta V/\Delta t$: Same notation for eq: 3 at line 191.

Done

Campi Flegrei is certainly one of the most studied volcanoes of the world. It is of course impossible to give proper credits to all previous work. There is however the need to cite some papers, but also to use their scientific content. Below some remarks/suggestions which can be useful for this study

Lines 27-30: “The mechanism generating inflation is still debated... omissis... plumbing system”. It should be made clear that what stated in text by Authors applies only to the current unrest, otherwise misconceptions and reasoning shortcuts keep propagating when they should not. In particular, the interpretation proposed in ref. 7 (plus other refs. below), based on geochemical data available until 2016, has its founding argument on the contrasting geochemical behaviors shown by the 1982-84 and post-2005 crises.

Note that in ref. 7, as well as in Moretti et al., 2013 EPSL), Moretti et al. 2018 (Sci. Reports) and Troise et al. (2019, ESR) the previous 1982-84 crisis is interpreted as strongly driven by the emplacement of a volatile-rich magma at 4 km depth, which leaves its residual of about 1 m at the end of the 1985-2004 subsidence (Moretti et al., 2018 Sci. Rep.; Troise et al., 2019 ESR). The contrasting geochemical behavior shown by the 1982-84 unrest vs the current one cannot be overlooked, because it is intrinsic to the data, is not derived from models or elaborations. It has profound implications, let's say an epistemic meaning, and despite being neglected by the literature cannot however be avoided. For more info please go through Moretti et al. (2020; Dev. in Volcanology book; Belkin, De Vivo, Rolandi eds.), who expand the reasoning introduced by Caliro et al. (2014, GCA). 2) Lines 33-34: Bevilacqua and coworkers have done the same and given values (Bevilacqua, A., Neri, A., De Martino, P., Isaia, R., Novellino, A., Tramparulo, F. D. A., & Vitale, S., 2020. Journal of Geodesy, 94, 1-27.). Please use it, at least to check

We specified we refer to the current unrest and we added the suggested references (Line 37).

Reviewer #3

This paper presents a model of evolving pressure, temperature, and gas contents in the shallow Campi Flegrei magma reservoir since 1950, to examine whether sufficient overpressure to induce eruptions is currently present or existed over the past decades of inflationary events. Quantifying conditions in the Campi Flegrei magma system is an important goal. However, I am unsure how one of the key figures in the paper (Fig. 2) is intended to be interpreted.

More than a focus on the overpressure within the magma reservoir, which is potentially affected by a multitude of factors for which we do not have extensive constraints, our contribution focuses on determining whether eruptible magma is currently present at shallow depth of about 4 km (As already stated in the abstract).

Additionally, some of the model result appear unintuitive, for example the observation that this model seems to allow continued intrusions of material with no net increases in overpressure, but with no stated mechanisms for viscous relaxation or fluid loss incorporated in the model. The authors provided minimal information about the model, either from the main text or the supplement. I am thus unsure how to evaluate

the model or understand the results. If this work is extended from previous models, then references should be clearly provided. If not, then much more detail is needed in this publication for readers to be able to interpret the results presented, given the potential societal significance of the claims the authors make about the current state of the volcano.

It seems that some portions of the text were unclear to the reviewer. We hope the revisions made it clearer. Our model does not consider the continuous injection of magma, but the injection in 4 main events associated with the 4 uplift events recorded since 1950 (this was stated at Lines 35-37 of the submitted manuscript and specified in the method section). Nevertheless, we added “instantaneous” referring to the injection episodes to clarify this potentially confusing point.

L10: comma not needed

Removed

Fig 2: It would be helpful here to add a subplot showing some of the data used to derive these model results, e.g., a timeseries of deformation data or inverted volume changes, or to add an earlier figure showing such data. Otherwise readers will have no sense of what the actual data looks like without going to other publications.

Fig 2: This figure is very difficult to interpret. It seems like the blue color is meant to map to the right (blue) axis, and the other colors map to the left axis. However, if that is the case, it is unclear why magma volume would decrease to zero very rapidly following injections of hotter magma but remain elevated for much longer following injections of cooler magma. The caption and/or figure needs to be modified to make clear exactly what is being plotted and what corresponds to what axes. If the blue lines that map to the left (black) axis are meant to represent magma volumes, it is then unclear how this relates to the other three excess fluid curves, if there are three distinct scenarios (with different injection temperatures) being shown. The supplement indirectly implies that these curves might instead represent the volume of magma within certain temperature bins, but nothing in the figure or caption clarifies this.

We are sorry the reviewer got confused. We have modified the caption as suggested to help the reader following the figure. Magma injection always occurred at the magma liquidus temperature, as specified in the originally submitted manuscript in the methods section. Additionally, Figure 2 was modified to include a legend as suggested by Reviewer#4.

Fig 3a: for the first injection, is it correct that the volume increase from second boiling is larger than the volume increase from the initial injection?

No as we simulated instantaneous injection. Also, Reviewer #1 was confused about this and therefore we reworked the corresponding portion of the text to clear up the confusion (From Line 157)

Fig 3b: why does compressibility appear to have an upper threshold at $\sim 3E-9$?

This is because we assume that the magma is a closed system and the flat portion of the line indicate that the entire magma volume has released all volatiles (i.e. magma at near solidus temperatures). We added this to the figure caption to clarify.

L111: Changing bulk magma compressibility will complicate things, but in general it seems that once a fluid phase is present the compressibility effects would not be able to indefinitely counter the effects of continued magma injection, and so pressure on the surrounding crust should begin increasing on average as magma input continues, since the authors state the model does not include fluid loss or viscous relaxation. This might relate to how the authors are treating solidified magma and its contributions to the total volume of crustal intrusion, but I am not clear on this point from the information provided.

As we model instantaneous injection, our statement here related more to the fact that the progressive increase of compressibility since 1950 makes the probability of an injection today to trigger an eruption lower than in the past. We modified the text to clarify this (Lines 175-177).

L142: This seems like an unnecessary level of approximation, it would be more meaningful, and seemingly relatively straightforward, to explicitly include viscous relaxation in the model.

We decided to use this approximation and calculate the relaxation timescale. We have tested the results also with the viscoelastic model of Jellinek and DePaolo, 2003, and we obtained the same overpressure. To avoid adding portion of text that are not extremely relevant for our target, we did not modify the text here.

L164: These seem like unrealistically small bounds on reservoir volume.

We are not sure on what this statement is based. This is one of the results of our thermal model and it corresponds to, approximately, the total injected volume (also indicating that only a small part of the magma has cooled to subsolidus temperatures).

L201: Fixed temperature boundary conditions would be most reasonable in the presence of relatively efficient hydrothermal circulation, is that the assumption here? *The temperature at the surface and the base of the simulation domain was considered fixed, not at the surface and base of the sill.*

L203: This statement seems to indicate that only one injection temperature is assumed, but how does this relate to the different temperatures in Fig 2?

Yes, in each of the 4 injection episodes, magma was injected at its liquidus temperature, so the statement is correct.

L207: Why these two choices, which are both relatively thin? It seems like a thicker sill would provide a better end-member estimate.

These are the most plausible thicknesses retrieved from geodetic data. We specified this in the text (Lines 87-92).

Thicker sills will result in slower cooling and more eruptible magma. We now specify this at Lines (Lines 100-102).

Reviewer #4

Thank you for the opportunity to read this comprehensive study on the evolution of unrest at Campi Flegrei caldera and its implications for future eruption scenarios. The paper addresses an important topic with significant implications for understanding volcanic systems and mitigating risks to the densely populated region surrounding the caldera. While the study provides valuable insights, there are several aspects that require clarification, additional validation, and further elaboration to strengthen the robustness and impact of your findings. The main conclusion of the paper is that while potentially eruptible magma has accumulated at a depth of approximately 4 km beneath Campi Flegrei caldera (Italy), the likelihood of an eruption being triggered solely by internal processes within the magma reservoir is currently low. This is due to several mitigating factors:

1. Small reservoir volume: The shallow magma reservoir is relatively small, which reduces the capacity for sufficient overpressure to trigger an eruption.
2. High magma compressibility: The presence of excess fluids increases magma compressibility, which dampens overpressure within the reservoir.
3. Viscous deformation of the crust: The surrounding crust's high temperature and low viscosity allow for deformation that accommodates reservoir expansion without generating critical overpressure.

The study indicates that continued magma injection at the current rate (8×10^6 m³/year) could lead to conditions favorable for an eruption within the next two decades. Additionally, the long-term integrity of the overlying crust is a critical factor. Fractures caused by decades of unrest, deformation, and hydrothermal activity could eventually form pathways for magma to reach the surface, potentially triggering an eruption even if the internal overpressure and reservoir size are suboptimal.

My main concern is about the uncertainty of the assumptions, for example:

The paper assumes a simplified geometry for the magma reservoir (e.g., thin sills at a fixed depth of 4 km). However, natural magmatic systems are often complex, with interconnected chambers and variable depths. How sensitive are the results to changes in the assumed geometry or depth of the reservoir?

We assume a sill-like shape for the shallow intrusion because of the results of geodetic inversion and because this geometry produces more conservative estimates in terms of eruptible magma present at shallow depth (i.e. a spherical or more vertically extended reservoir would cool less efficiently; This is now stated at Lines 87-102).

What is the uncertainty regarding the statement: "We assume a proportionality between the total measured uplift and the corresponding increase in volume estimated for the last two episodes (Table 1)." Especially, how to justify the diameters of the sills?

This assumption has been used in the past, as for eruptions for which geodetic inversion does not exist, there is nothing else we can do. We are looking for the results considering a scenario, we do not claim to simulate exactly the complexity of the magmatic system. This is clearly stated in the original manuscript (Lines 51-53). As explained above, the diameter of the sill we used is comparable to values obtained from geodetic inversion. Additionally, thermal modelling results (our model but also models from Annen and co-authors) show that as most of the heat is loss over the vertical direction, the thickness of the intrusion has a dominant control on cooling rates.

The thermal and mechanical models assume the reservoir behaves as a closed system for fluid release and pressure calculations. How realistic is this assumption, given that Campi Flegrei has a well-developed hydrothermal system that likely allows for fluid escape? How would incorporating open-system behavior affect the conclusions? *A similar comment was also made by Reviewer#2. We have now added to the text at Lines 54-59 and Lines 128-147 considerations about the hydrothermal system. In general terms, the escape of fluids will decrease the compressibility of magma within the reservoir, which we addressed already in the submitted manuscript in the text built around Figure 4.*

In Line 86: I don't quite understand: "However, even if half of the excess fluids would have been lost in the period 1950–1983, the excess fluid volume in the shallow reservoir would still be sufficient to explain the deflation episode (Fig. 2)." If the excess fluids were degassing during 1950–1983, should we observe deflation during that time, or was the inflation due to magma injection partially compensated by degassing? Also, can the degassing of the excess fluids explain the deflation observed during 1984–2005?

We agree with the reviewer, this sentence was unclear and we have modified it (Lines 128-132; deflation was observed after the 1950 and 1970 unrest episodes; DelGaudio et al., 2010; JVGR).

Why is it valid to use the rock properties at 3 km depth (Line 128) to represent the stiffness of the entire volcanic system? I believe using a numerical model that couples thermal and mechanical processes would be more convincing for explaining the relationship between deformation and overpressure evolution.

We took a conservative approach in the sense that the overpressures developed considering the physical properties of the host rocks at 3km depth should be larger than if the rocks were hotter, as it is likely that deeper within the plumbing system the temperatures are higher. A fully coupled thermomechanical model is out of the scope of our contribution.

Another concern is about the method part; it is hard to reproduce the model result due to lack of details. For example:

1. Which numerical method is used: finite volume or finite element? which code?
2. What is the geometry of the model?

We added all the details to the Methods section.

3. What is the initial geothermal gradient ($135^{\circ}\text{C}/\text{km}$)? While this gradient may be applicable to the rock above the magma reservoir, is it valid for the far field?

Yes, the gradient used is $135^{\circ}\text{C}/\text{km}$ as specified in the Methods Section. While it is likely that the geothermal gradient decreases with distance from the Campi Flegrei region, as most of the heat is lost from the top surface of the intrusions, as shown by our results and existing thermal model results, the impact of later variation of the thermal gradient should be insignificant.

Some minor comments:

Figure 2: Need a more detailed caption to describe which y-axes those curves belong to. Color-coding the shadows of different temperature cases with a sequential colormap to indicate the temperature from low to high would make the figure more readable.

Thanks for the suggestions, which we took on board and used to modify Figure 2 and the new Figure S1, showing the results for a thinner sill of 15 m.

Line 27: I suggest adding a time series plot of deformation to Figure 2.

We decided not to add this because the figure would be too busy and also because we consider instantaneous injection, which is likely not appropriate for the ongoing unrest, but was chosen to have first order estimates of the potential volume of eruptible magma present today below the centre of the inflation.

Please check the format of the values and their units. In some cases, there is a space between the value and the unit, while in others, there is none. Additionally, the unit is missing in at least one instance (Line 53).

Done

Dear Editor,

We have now addressed the well-placed comment of Reviewer #3 by performing additional thermal modelling calculations and restructuring the section that was at the core of the Reviewer #3 comment. Importantly, we use our thermal modelling results and an existing model (Townsend et al., 2019), which makes our calculations solid as based on previously published research. Critically, two recent studies focusing on the seismicity at Campi Flegrei, highlight the formation of extended fractures that could favour eruption even if the reservoir is not ideally configured for an internally triggered eruption. We have integrated these reference and discuss the implications of their main findings on the potential evolution of the ongoing volcanic crisis.

The other 3 reviewers were all satisfied with our previous revisions.

Our reply to the Reviewer comment are in *italic*.

Reviewer #3 (Remarks to the Author):

In my initial review, I asked how the authors' model accommodates continued magma intrusions with no net pressure increase but no means of releasing pressure via fluid loss or viscous relaxation. I do not think the authors have addressed this concern or provided sufficient information for readers to reproduce or understand their model. Given that the model is the main focus of the paper, I am still unable to effectively evaluate the paper.

In response to my question, the authors state that their model considers instantaneous rather than continuous magma intrusion. However, added material needs to be accommodated by the crust whether it is intruded continuously or in discrete events. The authors next state that they modified a sentence to no longer mention overpressure, which does not address my fundamental question about their model.

Very generally, if material is intruded into some location in the crust, the crust must deform to accommodate the added material. If there is no inelastic crustal deformation, then this crustal deformation should result in elastic stress changes and increased pressure on the intruded material. The distribution of stress in the surrounding crust and the solidifying intrusion might change in complicated ways during solidification, but stress changes should still be present in the solidified intrusion and crust, and should contribute to pressure in any remaining fluid and/or subsequent intrusions. The authors state that in their model "the material below each sill is advected downward" to make room for each sill, presumably invoking some sort of "artificial inelastic deformation event" since no additional details are provided. Even if this approach is justifiable in some limits, e.g., when sufficient time has passed for most stresses from the previous intrusion to have relaxed away inelastically, the authors do not provide such justifications, and this would still not explain how fluid overpressure is able to drop to zero between intrusions. The only equation provided for overpressure in the authors' model is the simple relation $dp = dV/(\beta * V)$. In this equation, dp can only approach zero if either dV approaches zero, which is not consistent with the authors' figures that show a finite volume of fluid persisting (even if the authors were to effectively justify neglecting solidified material), or if β or V approach infinity. The authors state that they treat the "the size of the magma reservoir as the calculated volume of magma above its solidus temperature", which does not appear to approach infinity from the provided plots. The authors also provide plots of β , showing maximum values of $\sim 3E-9$. So regardless of what the physical behavior

of the system should be, it is unclear how the authors' model produces the behavior they show based on the limited number of equations or other model descriptions provided.

The reviewer is correct, we wanted to simplify the pressurisation model as much as possible, but what we presented was indeed unclear. Thanks for reiterating your comments and for giving us the occasion to improve our manuscript.

During magma injection the overpressure within the reservoir does indeed increase. Because the current evolution of the overpressure within the magma reservoir is most relevant for the ongoing unrest, we decided to focus the section "Mechanisms of reservoir's overpressurisation and temporal evolution of overpressure" only on the last 10 years (i.e. the portion of the last unrest episodes during which magma injection has been suggested). First, we performed another simulation where we inject magma almost continuously (average rate of magma input of $8\text{Mm}^3/\text{y}$ distributed in single pulses every 0.1 years), starting from the thermal structure we calculated previously until 2015 (i.e. the beginning of the ongoing unrest crisis). We use our thermal model results to calculate the temporal evolution of the reservoir volume, and the volume of eruptible magma. To estimate whether the overpressure within the reservoir generated by magma input is potentially sufficient to lead to reservoir failure, we used the existing model of Townsend et al. (2019). To show the results of this analyses we changed Figure 3 completely.

We think this approach remove the ambiguity on the estimates of the reservoir overpressure while providing insights on potential of the shallow magma reservoir to feed volcanic activity. This also resulted in a shortening of this section, which became clearer and more streamlined.

Dear Editor,

We have addressed the last remaining comment of the Reviewer by adding error bars in Figure 3 and adding an explanation about the error bars in the main text (Lines 150-155) and in the caption of Figure 3.

Best Regards,

Luca Caricchi, corresponding author

**Scenario-based and physics-informed forecast of the evolution of 75 years of unrest at**
**Campi Flegrei caldera (Italy)**

Luca Caricchi¹, Charline Lormand¹, Stefano Carlino², Tommaso Pivetta², Guy Simpson¹

1) Department of Earth Sciences, University of Geneva, Geneva, Switzerland

2) INGV-Sezione di Napoli, Osservatorio Vesuviano, Via Diocleziano 328, 80124 Napoli, Italy

**Abstract**

**Campi Flegrei, which last erupted in 1538, has undergone unrest with periods of increased**
**seismicity, gas emission and ground deformation in the 50's, 70's 80's and since 2005. The eventual**
**culmination of this last episode in an eruption, will directly impact on 2 million people living in**
**the region, making it of critical concern for scientists, authorities and the general public. Here,**
**we use existing data, thermal modelling and calculations of the physical properties of magma, to**
**provide plausible future scenarios, under the assumption that magma injection at 4-5 km depth**
**is responsible for the unrest episodes recorded since 1950. We show that the ascent of magma and**
**magmatic fluids from depth is necessary to explain the measurements collected since 1950. Our**
**calculations suggest that potentially eruptible magma is present today at ~4 km depth, but that**
**eruption may be suppressed by the combination of small reservoir volume, magma**
**compressibility, and viscous deformation of the surrounding crust. We conclude that efforts**
**should focus on evaluation of the structural integrity of the crust overlying the shallow magma**
**reservoir.**

In the last 75 years the Campi Flegrei caldera (CFC) has experienced four episodes of unrest in 1950-
52, 1970-72, 1982-84, and the ongoing one that started in 2005^{1,2}. The surface uplift during these
episodes was 74, 160, 175 cm respectively and is larger than 130 cm at the moment of writing (Refs.³
and; Surveillance Bulletins of Istituto Nazionale di Geofisica e Vulcanologia, Osservatorio Vesuviano,
INGV-OV 2024). Between 1984 and 2005, the caldera floor subsided about 90 cm, thus recovering
about half of the inflation measured in the crisis of 1982-1984¹. The mechanism generating inflation is
still debated, with one possibility being the injection of magma at depth corresponding to the estimated

depth of the inflating source (4-5 km; Refs. ⁴⁻⁶) and another being an increased flux of magmatic fluids
released from deeper within the volcanic plumbing system⁷⁻¹⁰. Geodetic data for the 1982-1984 and the
ongoing unrest episodes are best explained by volume increase of a source located at 4 to 5 km depth
of $64-87 \times 10^6$ and $60 \times 10^6 \text{ m}^3$, respectively^{5,11}. Estimates for the 1950 and 1970 crises do not exist, thus
we assume a proportionality between the total measured uplift and the corresponding increase of volume
estimated for the last two episodes (Table 1).

Using the volumes obtained from geodetic inversion and estimated for the unrest episodes of the 50's
and 70's, we simulate an end-member extreme scenario where we assume that each inflation episode is
caused by the injection of magma. We perform thermal modelling simulating the injection of magma at
4km depth to trace the evolution of temperature, crystallinity, the capacity of magma to erupt
(eruptibility; magma is more likely to erupt if it contains less than 50 vol% crystals; Ref.¹²), excess fluid
fraction and magma compressibility, within this shallow reservoir from 1950 to today (Methods; Tables
1 and 2). Our simulations do not aim to reproduce the complex evolution of the unrest at CFC over the
last 75 years, but to provide first order estimates of the physical state of magma potentially present at
shallow depth and to trace the internal overpressure in this shallow reservoir. We use the deflation
between 1984 and 2005^{13,14} to cross-check if our calculations are appropriate, considering that the
calculated volume of excess fluids present within the reservoir in this time interval should be at least
sufficient to account for the volumetric decrease responsible for the deflation.

**Temporal evolution of crystallinity, volatile content and physical properties of magma injected in** 49 **the upper crust**

The CFC magma contains up to 4 wt.% of H₂O (Ref. ¹⁵) and becomes fluid-saturated at about 8 km
depth (~200 MPa), with variations related to CO₂ content of magma and excess fluids¹⁶. This depth
corresponds to the roof of the main magma reservoir, which develops mostly at depth > 8 km (Ref. ¹⁷).
Magma ascent from 8 to the depth at which the current deformation source is located (4 km equivalent
to ~100 MPa; Ref.^{5,18}), results in the degassing of about 1 wt.% H₂O followed by the release of
additional water during cooling and crystallisation (Fig. 1a). This implies that eventual magma
accumulating at 4 km depth is fluid saturated. The presence of an excess fluid phase increases magma

compressibility (Fig. 1b), which in turn, dampens the increase of overpressure within a magma reservoir
 caused by further exsolution of volatiles or the injection of additional magma^{19,20}.

Figure 1: Variation of melt fraction, excess fluid fraction (a) and magma compressibility (b) as function of temperature. These relationships were used to compute the temporal evolution of these parameters in time from the thermal modelling results.

Our results show that if magma was injected
 during each unrest episode that occurred over
 the last 75 years, the time interval between
 injections is not sufficient to allow for the
 temperature to return to the values preceding
 magma injection and the volume of magma
 above the solidus temperature increases with
 time (Fig. 2). Importantly, our calculations
 show that between 130×10^6 and 170×10^6 m³
 of eruptible magma are present today at 4 km
 depth (Table 2).

The cumulative volume of excess fluids in
 the reservoir at 4km depth, and therefore
 magma compressibility also increases
 progressively (Fig. 2; Table 2). The volume
 of excess fluids can be considered in relation
 to the deflation episode that took place

following the 1982-1984 crisis and lasted
 until 2005. In the period between 1984 and
 2005 the areas that uplifted by 175 cm in the
 1982-1984 unrest episode, underwent a
 deflation of 93 cm, thus recovering about half

of the inflation. The increase of volume for the 1982-1984 obtained from geodetic inversion was
 estimated to range between 64×10^6 and 87×10^6 m³ (Ref.¹¹). Thus, plausibly, the volume loss from
 1984 to 2005 should be half of the volume generating the inflation in 1982-1984 (i.e. $30 - 45 \times 10^6$ m³).
 Our calculations show that in the period corresponding to the deflation, about $70-95 \times 10^6$ m³ of excess

fluids would be present within the reservoir. These calculations consider the reservoir at 4 km depth as
 a closed system, which is unlikely. However, even if half of the excess fluids would have been lost in
 the period 1950-1983, the excess fluid volume in the shallow reservoir would still be sufficient to
 explain the deflation episode (Fig. 2), which indicates that our calculations are appropriate at least
 within an order of magnitude.

Figure 2: Temporal evolution of the volume of magma at different temperatures and excess fluids. The figure reports the results for the injections of 25m thickness and the maximum and minimum volume of the magma injection events (Table 1). The lower end of each coloured region is for the calculations performed considering the minimum volume of injected magma at each event.

Mechanisms of reservoir's overpressurisation and temporal evolution of overpressure

The increase of magma volume associated with the crystallisation-induced ("second boiling")
 exsolution of magmatic fluids can pressurise the magma reservoir, lead to inflation and if the pressure

reaches sufficiently high values²¹, trigger an eruption²²⁻²⁴. However, the release of fluids leads also to
 an increase of magma compressibility (Fig. 3b), which dampens the overpressure generated by a volume
 increase within the reservoir (Eq. 1; e.g. Ref. ²³):

$$98 \quad \Delta P = \frac{dV}{\beta V} \quad (1)$$

where ΔP is the overpressure, dV is the increase of volume and V is the volume of the reservoir and β
 is the compressibility of the magma within the reservoir.

Figure 3: Temporal evolution of the rate of volumetric variation, compressibility and overpressure. We plot two end member scenarios: in black the results obtained for injections of 15m thickness and the minimum volume for each injection, while in light-blue we show the results for injections of 25m thickness corresponding to the maximum volume estimated for each eruption. a) The increase of volume as function of time is the sum of the volume increase resulting from magma injection and the exsolution of fluids from the magma. The rate of volume increase is calculated for time intervals of 1 year. b) Evolution in time of the compressibility. c) Temporal evolution of the overpressure calculated using Equation 1.

To calculate the maximum overpressure generated by magmatic fluids exsolution, we assume the rocks
 surrounding the magma reservoir are fully elastic, despite this being highly unlikely given the high

geothermal gradient measured in boreholes²⁵. We compute the yearly rate of volume increase and
increase of magma compressibility resulting from fluid exsolution (Fig. 3a, b), and the size of the
magma reservoir as the calculated volume of magma above its solidus temperature. We use these values
to calculate the evolution of overpressure within the magma reservoir as function of time (Fig. 3c).
The results show the increase of the reservoir volume (Fig. 2), together with an increase of the volume
of excess fluids (i.e. magma compressibility; Figs. 1, 2), leads to a progressive decrease of the
overpressure generated by magma input and/or by fluid exsolution from 1950 to today (Fig. 3c). This
indicates that the probability of an eruption to be triggered by processes of overpressurisation internal
to the magma reservoirs decreases with time²⁴.

The results also show that the rates of volume increase generated by exsolution of magmatic fluids are
significantly lower than the rate of volume increase calculated with geodetic methods from 2007 to
today⁵ (Fig. 3a). This implies that the exsolution of volatiles from the shallow reservoir alone is not
sufficient to explain the ongoing inflation, suggesting, in agreement with Ref.⁵ that the input of magma
at 4 km depth is required to account for the ongoing inflation. Additionally, the gas emission monitoring
data suggest that the input of magma from depth is required both to account for the fluxes and
geochemistry of the released gases^{8,10}. The calculations show that overpressure compatible with
volcanic eruption develops only considering the injection of magma in the upper crust (Dashed vertical
lines in Fig. 3), while the exsolution of volatiles away from the magma injection period is not sufficient
to overpressurise the reservoir to critical values compatible with eruption (Fig. 3c).

The overpressure calculations we have performed with Equation 1 (Fig. 3c) are maximum estimates as
we considered the rock as fully elastic. However, because of the elevated temperature of the rocks
surrounding the reservoir at 4 km depth²⁵, the crust is not fully elastic, which makes the maximum
pressure generated by an increase of volume within the reservoir dependent on the viscosity of the
surrounding crust²¹. Beneath the volcanic island of Ischia, which is less than 20 km away from the
current unrest, the geothermal gradient is similar to that of CFC²⁵ and the viscosity of the crust (η_{crust}) at
3 km depth has been estimated from thermal modelling at about 2.5×10^{18} Pa s (Ref. ²⁶) and between 5

$\times 10^{16}$ Pa s and 2.7×10^{17} Pa s using the post-seismic relaxation time of 2017 Ischia earthquake (Pino et
al., 2023). Considering the crust as a Maxwell body, its relaxation timescale (τ) can be calculated as:

$$131 \quad \tau = \frac{\eta_{crust}}{G} \quad (2)$$

where G is the shear modulus²⁷ (3×10^{10} Pa). For timescales exceeding τ an increase of volume will not
generate additional overpressure but result in the accommodation of the volume increase by non-
recoverable deformation of the crust. Considering the values of 2.5×10^{18} Pa s, τ is 2.6 years. Taking
into account that the depth of magma injection at CFC is deeper than 3 km (Ref. ⁵) the temperature will
be higher and the viscosity lower than that estimated at Ischia for a depth of 3 km. For these reasons,
we consider the viscosity value around 10^{17} Pa s to be more appropriate, which results in a τ of 0.26
138 years (Eq. 2). Considering the maximum rate of magma input, recorded since 2022, of 8.7×10^6 m³/y
estimated by Ref. ⁵, and a relaxation timescale of 0.26 years, we can calculate the current maximum
overpressure as:

$$141 \quad \Delta P = \frac{dV_{0.26}}{\beta V_r} \quad (3)$$

where $dV_{0.26}$ is the volume increase over 0.26 years (i.e. 8×10^6 m³/y \times 0.26 y = 2.08×10^6 m³) and V_r is
the current volume of the magma reservoir calculated with thermal modelling (Methods). The resulting
values calculated for thickness of the injected magma sills of 15 and 25 m and the minimum and
maximum volumes estimated from geodetic methods vary from 4 MPa to 6 MPa (Table 2). These values
are about 2 times smaller than the estimated value required to feed a volcanic eruption of silicic
magma^{21,24,28,29}. This estimate assumes that no gas was released from the reservoir in the last 75 years
and thus considers the highest possible compressibility. We calculate the increase of compressibility
due to a drop of the excess fluid fraction of 50% relative. Our results show that the average excess fluid
fraction at the end of our calculations reaches a value of about 0.22 (Fig. 4a and Table 2; calculated for
a thickness of the injected sills of 15 m and the minimum injected volume, which maximises the fraction
of excess fluid released because of the highest rates of cooling of these low aspect ratio sills). The
removal of half of the excess fluids, results in a drop of compressibility of about 50%. Thus, considering
the release of 50% of the excess fluids from 1950 to today and the resulting 50% decrease in

compressibility (Eq.3; Fig. 4), the overpressure generated by the injection of magma at a rate of 8×10^6
 m^3/y would reach values of 8 to 12 MPa, which is close to **critical values** compatible with eruption^{21,28,29}.

Figure 4: Temporal evolution of the volume fraction of excess fluids and its relationship with magma compressibility. (a) Evolution in time of the average fraction of excess fluids present within the magma reservoir as calculated from the thermal model performed considering sills of 15m thickness and a volume of injected magma corresponding to the minimum estimated volume injection for each episode. (b) Relationships between the average compressibility of magma within the reservoir and the average volume fraction of excess fluids present within the reservoir. The dashed line in panel a and b show the value of compressibility calculated at the end of the simulation. The black line in panel b shows the drop of compressibility resulting by the removal of about 50% of the exsolved fluids.

Our calculations show that values of overpressure
 above 10 MPa were potentially exceeded in the past (Fig. 3c) but no eruption occurred since 1538. It is
 important to notice that overcoming the overpressure required to fracture the rocks surrounding a
 magma reservoir is not sufficient to allow the magma to reach the surface and feed a volcanic eruption
 when considering internal processes of reservoir pressurisation²⁹. Numerical modelling shows that a
 magma reservoir should also have a minimum volume to feed a volcanic eruption and that this volume
 increases with the depth of pre-eruptive magma storage³⁰. Our thermal model results provide estimates
 of the current volume of the magma reservoir that vary between 0.21 and 0.35 km^3 (Table 2), which is
 about half of the size of the volume of the reservoir required to feed an eruption of the magnitude of

[revised manuscript text omitted]

- 5. Astort, A. *et al.* Tracking the 2007–2023 magma-driven unrest at Campi Flegrei caldera (Italy).
 *Commun Earth Environ* **5**, 506 (2024).
 6. D’Auria, L. *et al.* Magma injection beneath the urban area of Naples: a new mechanism for the
 2012–2013 volcanic unrest at Campi Flegrei caldera. *Sci Rep* **5**, 13100 (2015).
 7. Moretti, R., De Natale, G. & Troise, C. A geochemical and geophysical reappraisal to the
 significance of the recent unrest at Campi Flegrei caldera (Southern Italy). *Geochemistry,*
 *Geophysics, Geosystems* **18**, 1244–1269 (2017).
 8. Chiodini, G. *et al.* Hydrothermal pressure-temperature control on CO₂ emissions and
 seismicity at Campi Flegrei (Italy). *Journal of Volcanology and Geothermal Research* **414**,
 107245 (2021).
 9. Buono, G. *et al.* New Insights Into the Recent Magma Dynamics Under Campi Flegrei Caldera
 (Italy) From Petrological and Geochemical Evidence. *J Geophys Res Solid Earth* **127**, (2022).
 10. Caliro, S. *et al.* Escalation of caldera unrest indicated by increasing emission of isotopically
 light sulfur. *Nat Geosci* (2025) doi:10.1038/s41561-024-01632-w.
 11. Gottsmann, J., Folch, A. & Rymer, H. Unrest at Campi Flegrei: A contribution to the magmatic
 versus hydrothermal debate from inverse and finite element modeling. *J Geophys Res* **111**,
 B07203 (2006).
 12. Takeuchi, S. Preeruptive magma viscosity: An important measure of magma eruptibility. *J*
 *Geophys Res* **116**, B10201 (2011).
 13. Troiano, A., Di Giuseppe, M. G., Patella, D., Troise, C. & De Natale, G. Electromagnetic
 outline of the Solfatara–Pisciarelli hydrothermal system, Campi Flegrei (Southern Italy).
 *Journal of Volcanology and Geothermal Research* **277**, 9–21 (2014).
 14. Troise, C., De Natale, G., Schiavone, R., Somma, R. & Moretti, R. The Campi Flegrei caldera
 unrest: Discriminating magma intrusions from hydrothermal effects and implications for
 possible evolution. *Earth Sci Rev* **188**, 108–122 (2019).
 15. Stock, M. J. *et al.* Tracking volatile behaviour in sub-volcanic plumbing systems using apatite
 and glass: Insights into pre-eruptive processes at Campi Flegrei, Italy. *Journal of Petrology* **59**,
 2463–2492 (2018).
 16. Vetere, F., Botcharnikov, R. E., Holtz, F., Behrens, H. & De Rosa, R. Solubility of H₂O and
 CO₂ in shoshonitic melts at 1250°C and pressures from 50 to 400MPa: Implications for Campi
 Flegrei magmatic systems. *Journal of Volcanology and Geothermal Research* **202**, 251–261
 (2011).
 17. Zollo, A. *et al.* Seismic reflections reveal a massive melt layer feeding Campi Flegrei caldera.
 *Geophys Res Lett* **35**, n/a-n/a (2008).
 18. Cubellis, E., Luongo, G., Obrizzo, F. & Sepe, V. *Contribution to Knowledge Regarding the*
 *Sources of Earthquakes on the Island of Ischia (Southern Italy). Natural Hazards* (Springer
 Netherlands, 2020). doi:10.1007/s11069-019-03833-8.
 19. Kilbride, B. M., Edmonds, M. & Biggs, J. Observing eruptions of gas-rich compressible
 magmas from space. *Nat Commun* **7**, 13744 (2016).
 20. Edmonds, M. & Woods, A. W. Exsolved volatiles in magma reservoirs. *Journal of Volcanology*
 *and Geothermal Research* **368**, 13–30 (2018).
 21. Jellinek, A. M. & DePaolo, D. J. A model for the origin of large silicic magma chambers:
 precursors of caldera-forming eruptions. *Bull Volcanol* **65**, 363–381 (2003).
 22. Tait, S., Jaupart, C. & VERGNOLLE, S. Pressure, Gas Content and Eruption Periodicity of a
 Shallow, Crystallizing Magma Chamber. *Earth Planet Sci Lett* **92**, 107–123 (1989).
 23. Tramontano, S., Gualda, G. A. R. & Ghiorso, M. S. Internal triggering of volcanic eruptions:
 tracking overpressure regimes for giant magma bodies. *Earth Planet Sci Lett* **472**, 142–151
 (2017).
 24. Caricchi, L., Townsend, M., Rivalta, E. & Namiki, A. The build-up and triggers of volcanic
 eruptions. *Nat Rev Earth Environ* **2**, 458–476 (2021).
 25. Carlino, S. Heat flow and geothermal gradients of the Campania region (Southern Italy) and
 their relationship to volcanism and tectonics. *Journal of Volcanology and Geothermal*
 *Research* **365**, 23–37 (2018).

- 26. Castaldo, R. *et al.* The role of thermo-rheological properties of the crust beneath Ischia Island
(Southern Italy) in the modulation of the ground deformation pattern. *Journal of Volcanology*
*and Geothermal Research* **344**, 154–173 (2017).
- 27. Ji, S., Sun, S., Wang, Q. & Marcotte, D. Lamé parameters of common rocks in the Earth's
crust and upper mantle. *J Geophys Res Solid Earth* **115**, (2010).
- 28. Rubin, A. M. Propagation of magma-filled cracks. *Annu Rev Earth Planet Sci* (1995)
doi:0084-6597/95/0515-0287.
- 29. Rubin, A. M. Getting Granite Dikes Out of the Source Region. *Journal of Geophysical*
*Research-Solid Earth* **100**, 5911–5929 (1995).
- 30. Townsend, M. & Huber, C. A critical magma chamber size for volcanic eruptions. *Geology* **48**,
431–435 (2020).
- 31. Huppert, H. E. & Woods, A. W. The role of volatiles in magma chamber dynamics. *Nature*
**420**, 493–495 (2002).
- 32. Biggs, J. *et al.* Fracturing around magma reservoirs can explain variations in surface uplift
rates even at constant volumetric flux. *Journal of Volcanology and Geothermal Research* **452**,
108129 (2024).
- 33. Marsh, B. D. On the Crystallinity, Probability of Occurrence, and Rheology of Lava and
Magma. *Contribution to mineralogy and petrology* **78**, 85–98 (1981).
- 34. Pereira, M. L., Zanon, V., Fernandes, I., Pappalardo, L. & Viveiros, F. Hydrothermal alteration
and physical and mechanical properties of rocks in a volcanic environment: A review. *Earth*
*Sci Rev* **252**, 104754 (2024).
- 35. Heap, M. J., Baud, P., Meredith, P. G., Vinciguerra, S. & Reuschlé, T. The permeability and
elastic moduli of tuff from Campi Flegrei, Italy: implications for ground deformation
modelling. *Solid Earth* **5**, 25–44 (2014).
- 36. Fowler, S. J., Spera, F. J., bohrson, W. A., Belkin, H. E. & De Vivo, B. Phase Equilibria
Constraints on the Chemical and Physical Evolution of the Campanian Ignimbrite. *Journal of*
*Petrology* **48**, 459–493 (2006).

Table 1: Estimates of volumetric variations associated with th

Unrest period	$\Delta V \text{ m}^3 \times 10^6$	Sill diameter (m) for h=15 m
1950-1952	26 - 36	742 - 874
1970-1972	60 - 82	1128 - 1319
1982-1984	64 - 87	1165 - 1359
2007-2023	60	1128

Figure 4 unrest episodes, and diameters of the sill used for thermal modelling

Sill diameter (m) for h=25 m
575 - 677
874 - 1021
902 - 1052
874

Table 2: Calculated current volume of the magma reservoir, eruptible volume

Thickness of injections (m)	Vol injections (km³) x 10⁻²	Vol. reservoir (km³)
15	26 - 60 - 64 - 60	0.21
15	36 - 82 - 87 - 60	0.35
25	26 - 60 - 64 - 60	0.28
25	36 - 82 - 87 - 60	0.28

α, volume fraction of excess fluids and overpressure that would be ge

Vol. eruptible (km ³)	Vol. Fraction exsolved fluids (closed system)
0.16	0.24
0.16	0.21
0.14	0.21
0.13	0.21

generated by the injection at the current inverted rates of $8 \times 10^6 \text{ m}^3/\text{y}$ for different model c

Overpressure (MPa)
5.92
3.98
5.22
5.14

onfigurations